# Domain Adaptation with Invariant Representation Learning: What Transformations to Learn?

**Petar Stojanov**[1,4*], **Zijian Li**[2*], **Mingming Gong**[3],
**Ruichu Cai**[2], **Jaime G. Carbonell**[1], **Kun Zhang**[1]
[1] Carnegie Mellon University
[2] School of Computer Science, Guangdong University of Technology
[3] School of Mathematics and Statistics, University of Melbourne
[4] Broad Institute of MIT and Harvard

## Abstract

Unsupervised domain adaptation, as a prevalent transfer learning setting, spans many real-world applications. With the increasing representational power and applicability of neural networks, state-of-the-art domain adaptation methods make use of deep architectures to map the input features $X$ to a latent representation $Z$ that has the same marginal distribution across domains. This has been shown to be insufficient for generating optimal representation for classification, and to find conditionally invariant representations, usually strong assumptions are needed. We provide reasoning why when the supports of the source and target data from overlap, any map of $X$ that is fixed across domains may not be suitable for domain adaptation via invariant features. Furthermore, we develop an efficient technique in which the optimal map from $X$ to $Z$ also takes domain-specific information as input, in addition to the features $X$. By using the property of minimal changes of causal mechanisms across domains, our model also takes into account the domain-specific information to ensure that the latent representation $Z$ does not discard valuable information about $Y$. We demonstrate the efficacy of our method via synthetic and real-world data experiments. The code is available at: https://github.com/DMIRLAB-Group/DSAN.

## 1 Introduction

Unsupervised domain adaptation (UDA) is a common setting for supervised learning, in which the labeled training and unlabeled test data come from different distributions. More formally, given features $X \in \mathbb{R}^d$ and labels $Y \in \mathbb{R}$, we observe labeled source and unlabeled target domain instances, represented by $(\mathbf{x}^\mathcal{S}, \mathbf{y}^\mathcal{S}) = (\mathbf{x}_k^\mathcal{S}, y^\mathcal{S})_{k=1}^{m_S}$ and $\mathbf{x}_k^\mathcal{T} = (\mathbf{x}^\mathcal{T})_{k=1}^{m_T}$ respectively, where $P^\mathcal{S}(X, Y) \neq P^\mathcal{T}(X, Y)$, and $m_S$ and $m_T$ are the number of observations in the source and target domains, respectively. The main challenge of domain adaptation is to use the given source domain observations for learning a predictor that will perform well in the target domain. To do so, the procedure needs to make use of some similarities between the two domains. One way to formalize this is by making assumptions about how the joint distribution joint $P(X, Y)$ changes across domains. For example, in the well-studied setting of covariate shift [31, 44, 20, 35, 2, 8], the marginal distribution $P(X)$ changes while the conditional distribution $P(Y|X)$ (i.e. the optimal predictor) is shared across domains.

However, in many real-world applications, $P(Y|X)$ can also change, and this requires making use of further assumptions. One such assumption is that the factorization $P(X, Y) = P(Y)P(X|Y)$ allows for addressing the changes in $P(Y)$ and $P(X|Y)$ independently in a situation where their

---

*These authors contributed equally to this work.

35th Conference on Neural Information Processing Systems (NeurIPS 2021).

respective changes are simple and easier to capture [48, 30]. In this particular setting, the problem is generally broken down into: **(1)** *Target shift:* $P(Y)$ changes across domains while $P(X|Y)$ stay the same [34, 21, 29, 48] **(2)** *Conditional shift:* $P(X|Y)$ changes across domains but $P(Y)$ stays the same [48, 26, 24]. **(3)** *Conditional-target shift:* Both $P(X|Y)$ and $P(Y)$ change independently across domains - the most general setting under this generating process assumption [48, 6]. Then, assumptions about the changes of the factors of the joint distrubution can be made so that the problem is solvable, such as location-scale transformation [48, 26], or that the changing parameters lie on a low-dimensional manifold [33], and algorithms can be designed to enforce these constraints and make use of them for prediction in the target domain.

Another fruitful view of the problem is through the lens of representation learning, due to wide-spread applicability of neural architectures for many real-world problems. In particular, state of the art deep learning techniques harness the high representational capacity of neural networks to transform the input data into a latent feature representation which is predictive of the target variable $Y$ in the source domain, and has the same distribution across domains. Formally, this means learning a function (encoder) $\phi : \mathcal{X} \to \mathcal{Z}$ from the input space to a latent space $\mathcal{Z}$, such that $P^S(Z) = P^T(Z)$. At the same time, a function $h : \mathcal{Z} \to \mathcal{Y}$ can be learnt to minimize the risk in the labeled source domain [1]. The hope is then, that the overall function $g := h \circ \phi$ will have low prediction risk in the target domain. However, the above-described theoretical and methodological framework does not guarantee that the learnt representation $Z$ will have any relevant information for predicting $Y$ in the target domain. Namely, one can easily have a situation in which the learnt representation $Z$ is *marginally invariant* ($P^S(Z) = P^T(Z)$), but not *conditionally invariant* ($P^S(Z|Y) \neq P^T(Z|Y)$), as discussed in [53]. This means that the learnt function $g := h \circ \phi$ can have very good prediction performance in the source domain, but generalize very poorly to the target domain.

In many methods, the same encoding function $\phi(X)$ across domains is used to learn invariant latent representations. This enjoys computational benefits and makes the learning procedure relatively simple, and the vast majority of approaches ([12, 22, 25, 23, 17] among many) employ this technique. However, in certain situations, the same encoding function across domains cannot learn a marginally invariant representation that is optimal for classification in the target domain. There are certain studies ([37, 38, 4]) which implement domain-specific encoders $\phi_S$ and $\phi_T$. Unfortunately, such methods suffer from the following drawbacks: **(1)** the exact motivation behind having separate encoders for each domain is not clear; **(2)** including two separate encoders may be inefficient because it greatly increases the number of parameters that we need to learn; **(3)** in the field of UDA based on invariant

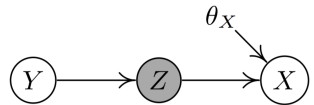

Figure 1: The underlying data-generating process under conditional shift, of the observed variables $Y$ and $X$, and the latent variable $Z$. $\theta_X$ represents the changing parameters of $P(X|Y)$ across domains.

representation learning, there is still no principled way to guarantee that the learned marginally invariant representation $Z$ has sufficient structural (semantic) information, and has the potential to be conditionally invariant.

In this paper we assume the setting of *conditional shift*, and we make use of the data-generating process to: **(i)** justify the use of two separate encoding functions in order to infer the latent representation, **(ii)** implement the two encoding functions more efficiently, and **(iii)** constrain the latent representation $Z$ to have meaningful structure which is useful for prediction in the target domain. In Section 2, we first motivate the use of two separate encoders to infer $Z$, via rigorous treatment and an illustrative example. Subsequently, in Chapter 3, we introduce an efficient way to implement two separate functions for inferring $Z$. In Chapter 3 we shall also introduce a principled way to ensure that the latent representation $Z$ contains useful information for prediction, and finally, in Chapter 4 we provide empirical evaluation.

## 1.1 Related Work

As previously mentioned, there is a vast body of work exploring approaches to make use of latent invariant representations for the purposes of unsupervised domain adaptation, using both deep learning and more traditional techniques. Namely, invariant representations were considered as a linear projection in [26], where theoretical guarantees were established regarding the conditional

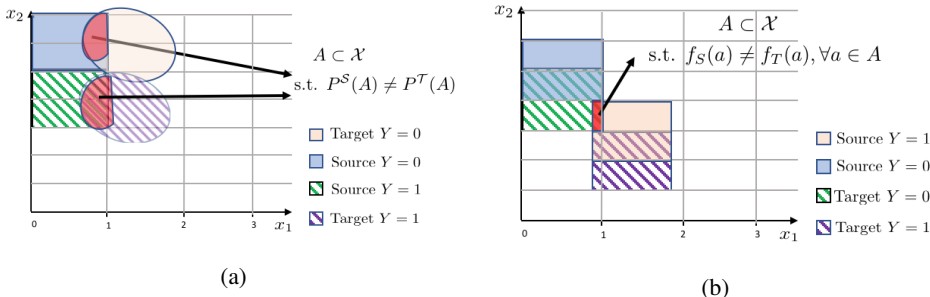

Figure 2: Two different scenarios in which a single encoder $\phi(X)$ is not sufficient to learn $Z$.

invariance of the latent representation and the identifiability of the joint distribution in the target domain. Furthermore, this framework was generalized to nonlinear maps to invariant latent representations, parameterized by neural network termed DANN [12]. In this framework, the constraint that $P^S(Z) = P^T(Z)$ can be enforced in various ways, including adversarial training, Maximum Mean Discrepancy [22], Wassertein Distance [9] and Margin Disparity Discrepancy [51]. However, none of these studies have considered using two separate encoders for the two domains.

Subsequently, this reasoning has been considered in combination with image translation (CYCADA) [19]. Another assumption frequently made in recent studies is that the input features $X$ have a clustering structure in which each cluster has a different label $Y$. Therefore, further directions were pursued by incorporating additional information contained in the unlabeled target domain data, such as pseudo-labels provided by the classifier which is initially trained on the labeled source domain data [23, 41] (termed CDAN and SLPP, respectively). In addition, one can also make use of this assumption to enforce that the prediction function $g$ does not pass through high-density regions in input space, as performed by the DIRT-T algorithm [32]. The method by [37] makes use of two separate transformation functions, but does not regularize their output to preserve semantic information in any way. In addition to these, approaches, the DSN algorithm [4] takes into models the changing parameters of $P(X, Y)$ in the latent representation and enforces that they are different from the invariant representation Z. The only method that we are aware of which takes domain-specific information into account is the study by [38], in which domain information is given and continuous, and therefore contains a lot more information than the discrete domain index that is typically considered. In [17], a spherical architecture is used for $\phi$, however there is only one encoder for both domains. The study by [27] performs bridging between the source and the target domains using data-augmentation, but this is a technique specific to image data.

In addition to domain adaptation methodologies in the setting where the is only one labeled source domain, there have been several efforts to tackle the problem of multiple-source unsupervised domain adaptation. Namely, the method by [3] tackles the problem by using modified kernel SVMs to operate on both input features $X$ and marginal distributions $P_X$. Furthermore, the method by [46] assumes that the target domain is a linear mixture of the source domains, and infers the linear coefficients and makes use of them for prediction in the target domain. Furthermore, in [47], data-generating process information is discovered from the multiple-domain data, and made use for recovering the changing parameters in the target domain in order to construct a predictor for it.

While the above-mentioned methods use various approaches to improve the quality of the learnt invariant latent representation and show competitive performance, there is no guarantee that these approaches are general enough to encompass all scenarios. Namely, the challenge that existing methods face is two-fold: **(1)** there is no principled way to ensure that marginal invariance $(P^S(Z) = P^T(Z))$ preserves conditional invariance $(P^S(Z|Y) = P^T(Z|Y))$, **(2)** there are cases (especially in lower-dimensional datasets) where the encoding function $\phi$ needs domain-specific information, in addition to the features $X$. The aim of this paper is to address these two issues. In this paper, we introduce the first invariant representation learning method that makes use of the data-generating process to justify and efficiently use domain-specific information in the encoder $\phi$, and to properly constrain it to ensure that $Z$ contains relevant label-specific information.

## 2 What Does it Take to Learn a Useful Invariant Representation?

To gain insights into what information is necessary for learning an intermediate representation $Z$ via $\phi$, let us consider the data-generating process depicted on Figure 1. In this graphical model, the label $Y$ is generated first from its prior distribution $P(Y)$. Then, the invariant representation $Z$ is generated from $Y$ via the generating mechanism $P(Z|Y)$ and $X$ is subsequently generated from $P(X|Z; \theta_X)$, where $\theta_X$ are the changing parameters of $P(X|Y)$ across domains. We consider $\theta_X$ as a parameter vector with a constant value for each point within a domain, and we generally assume that $\theta_X$ contains little information, i.e. the change across domains in $P(X|Y)$ is *minimal*. This generating process, in part or in whole, can be exploited in many classification applications. For example, in object classification in images, $Y$ is the label (the object the picture was taken of), $X$ is the observed pixels, and $Z$ can correspond to latent features with semantic relevance, such as some function of the edges and contours on the image. In this case, $\theta_X$ can correspond to different resolution, illumination conditions, or other environment-specific changes that are not relevant for predicting the class $Y$. In this graphical representation, we see that generally speaking, $Z$ is conditionally dependent from $\theta_X$ given $X$, although they may be marginally independent. This implies that in order to recover $Z$, given that $X$ is considered, the information of $\theta_X$ should also be considered in the transformation, because it is in the Markov Blanket of $Z$ (the Markov Blanket of a variable of interest consists of all other variables that are either its parents, its children, or the parents of its children in the probabilistic graphical model. The Markov Blanket represents all of the information needed to predict the variable of interest).

To gain intuition why considering the Markov Blanket is important, consider the following scenario, depicted in Figure 2a. In this setting, we are given the source domain with Uniform distribution, and the target domain with Gaussian distribution. Here, we have a region $A \subset \mathcal{X}$ (highlighted in red), in which both of the domains' distributions have support, so there is support overlap. However, each domain has a different density in this region. The following proposition claims that in this situation, we cannot infer an invariant representation $Z$ from $X$ with a single encoder $\phi$:

**Proposition 1:** *Let $A \subset \mathcal{X}$ be a region in input space such that $P^{\mathcal{S}}(X \in A) > 0$ and $P^{\mathcal{T}}(X \in A) > 0$, and $P^{\mathcal{S}}(X \in A) \neq P^{\mathcal{T}}(X \in A)$. Furthermore, let $\phi : \mathcal{X} \to \mathcal{Z}$ be an encoder s.t. $A = \{a : \phi(a) \in B\}$ for some $B \subset \mathcal{Z}$. Then, there is no function $\phi$ s.t. $P^{\mathcal{S}}(\phi(X) \in B) = P^{\mathcal{T}}(\phi(X) \in B)$.*

All omitted proofs can be found in the supplementary materials. This proposition implies that if the supports of the marginal distributions of the source and target domains overlap, and if the distributions are different in the specific region of the support overlap (set $A$), then we cannot use a fixed transformation to transform the data in that region to an invariant representation in $\mathcal{Z}$ space.

There is an additional case in which a single encoder $\phi$ is not sufficient to learn an optimal latent representation $Z$. The scenario is illustrated on Figure 2b. In this case, even though both domains have the same distribution $P(X)$ in the region of support overlap between the source and the target domain (labeled in red), the value of $Y$ is opposite between the two domains. The following proposition demonstrates that in this setting, a single encoding function $\phi$ is inherently sub-optimal:

**Proposition 2:** *Let the true labeling functions in the source and target domain be $f_S, f_T : \mathcal{X} \to \mathcal{Y}$, respectively. Let $A \subset \mathcal{X}$ be a region s.t. $f_S(a) \neq f_T(a), \forall a \in A$. Let $g : \mathcal{X} \to \mathcal{Y}$ be a composition of a representation learner $\phi : \mathcal{X} \to \mathcal{Z}$ and a classifier $h : \mathcal{Z} \to \mathcal{Y}$. If $\phi$ is the same function across domains, then for a 0-1 loss, the risk over the region $A$: $\epsilon^A(g) = \epsilon_{\mathcal{S}}^A(g) + \epsilon_{\mathcal{T}}^A(g) \geq 1$, where $\epsilon_{\mathcal{S}}^A(g)$ and $\epsilon_{\mathcal{T}}^A(g)$ are the source and target domain risks respectively.*

In this scenario, no matter what criterion one uses to find the representation for downstream classification using $h$, as long as the two domains use the same transformation $\phi$ for the representation, the points in $A$ cannot be correctly classified in the target domain, if they are correctly classfied in the source domain. This is because the final classifer of the data, $g = h \circ \phi$, is fixed across domains. As a consequence, any point $a \in A$ will be mapped to the same class label, no matter which domain it is in. These two scenarios generally occur in low-dimensional datasets, such as scientific and healthcare applications. In very high-dimensional structured data such as images and text, there will

be some domain-specific dimensions which will have very different values in the source and the target domain (such as brightness in images in daytime vs. nighttime). This will result in no overlap between the supports of the source and target domain distributions, and thus the domain-specific dimensions will act as domain specific information. In this case using $\phi$ which is only a function of $X$ is sufficient. However, even in this scenario, we need to find a principled way to constrain the encoder such that $Z$ contains relevant label-specific information for the prediction in the target domain.

The observations from the perspective of the causal mechanism of generating $X$ from $Z$ and $\theta_X$ provide hints how to tackle the problem of UDA using invariant latent representations in a general way. First of all, we observe that $\theta_X$ needs to be an input to the encoder $\phi$, in addition to X. Secondly, in this causal mechanism, we can safely assume that the influence of $\theta_X$ on the relationship between $X$ and $Z$ (and therefore the relationship between $X$ and $Y$) is minimal. This allows us to: **(1)** model latent variable $\theta_X$ as a function of the domain index and provide it is an input to $\phi(X, \theta_X)$, instead of learning two separate encoders $\phi_\mathcal{S}$ and $\phi_\mathcal{T}$; **(2)** mimic the causal mechanism of generating X, given by $X = \tilde{\phi}(Z, \theta_X)$, via a decoder $\tilde{\phi} : \mathcal{Z} \times \Theta \to \mathcal{X}$, in which we constrain the influence of $\theta_X$ to be *minimal*. The hope is then that such a decoder would act as a regularizer on the encoder $\phi$, forcing it to preserve important semantic information when inferring $Z$. In this paper, we design a method based on this reasoning in order to address the problem of UDA in a principled manner.

## 3 Proposed Method: Domain-Specific Adversarial Training

Motivated by the above discussion, we aim to design a domain-adversarial network which can make use of the domain-specific parameters $\theta_X$. Let $\theta_X \in \{\theta_X^\mathcal{S}, \theta_X^\mathcal{T}\}$, which are its possible values in the source and target domains. The model for the proposed method DSAN (Domain-Specific Adversarial Network) is depicted in Figure 3.

We first consider the domain index of the data: $j \in \{\mathcal{S}, \mathcal{T}\}$ (which indicates whether a data-point comes from the source or target domain), and parameterize the estimated latent domain-specific parameters $\hat{\theta}_X^{(j)}$ as a function of the domain index of the $i$-the data-point $j_i$ (0 for source, 1 for target domain). We make use of $\hat{\theta}_X^{(j)}$ and data encoder $\phi$ to obtain $\mathbf{z}_i^{(j)} = \phi(\mathbf{x}_i^{(j)}, \hat{\theta}_X^{(j)})$, where $\mathbf{x}_i^{(j)}$ and $\mathbf{z}_i^{(j)}$ represent the the $i$-th input point and latent representation point respectively, from the $j$-th domain. We can then use the latent representation $\mathbf{z}_i^{(j)}$ to obtain a softmax probability for the predicted label label $\tilde{y}_i^{(j)} = h(\mathbf{z}_i^{(j)})$, and this predictor can be trained in the source domain. To enforce the condition $P^\mathcal{S}(Z) = P^\mathcal{T}(Z)$, we also have an adversarial predictor which predicts the domain index, given by $\tilde{j} = h_a(\mathbf{z}_i^{(j)})$. In order to constrain the influence of domain-specific changes $\theta_X$, our model also needs to make use of (or mimic) the data-generating process depicted on Figure 1, in particular the process of generating $X$ from $Z$ and $\theta_X$. This can be achieved via a decoder $\tilde{\phi}$ which reconstructs the features $X$ in both the source and the target domain, from $Z$ and $\theta_X$, given by $\tilde{\mathbf{x}}_i^{(j)} = \tilde{\phi}(\mathbf{z}_i^{(j)}, \hat{\theta}_X^{(j)})$. As described so far, our model consists of the following loss functions:

$$\mathcal{L}_{reconst.} = \frac{1}{m_S + m_T} \sum_{i=1}^{m_S+m_T} ||\mathbf{x}_i - \tilde{\mathbf{x}}||_2^2, \quad \mathcal{L}_{classif.} = -\frac{1}{m_S} \sum_{c=1}^{C} \sum_{i=1}^{m_S} y_{ic} log(\tilde{y}_{ic}),$$

$$\mathcal{L}_{inv.} = -\sum_{i=1}^{m_S+m_T} \{j_i \log \tilde{j}_i + (1-j_i) \log(1-\tilde{j}_i)\}, \quad \mathcal{L}_{cent.} = -\sum_{i=1}^{m_T} h(\mathbf{z}_i^\mathcal{T})^T \log(h(\mathbf{z}_i^\mathcal{T}))$$

Here, $\mathcal{L}_{reconst.}$, $\mathcal{L}_{classif.}$, and $\mathcal{L}_{inv.}$ are the reconstruction, classification and adversarial loss respectively. $\mathcal{L}_{cent.}$ is the conditional cross-entropy of the predictions in the target domain, and intuitively, it serves to enforce that the decision boundary does not cross data-dense regions [32]. The current loss functions do not involve a constraint on the encoding transformation $\phi$ which would prevent it from discarding valuable information about $Y$. We now describe how we can make use of the data-generating process and the assumption of minimal change across domains to enforce such a constraint.

### 3.1 Enforcing Minimal Change by Mutual Information Minimization

Before we delve into the details of how to enforce minimality of the influence of $\theta_X$ on the generating process of $X$ from $Y$, we need to find a way to measure it. One way to formally do so is the joint

mutual information $I(\theta_X; (X, Y))$. Conveniently, this term is related to a distributional divergence measure that can be minimized, as stated by the following lemma:

**Lemma 1:** *Let $X, Y \sim P(X, Y; \theta_X)$, and let $\theta_X$ be a discrete random variable with possible values $\theta_X^{\mathcal{S}}$ and $\theta_X^{\mathcal{S}}$ in the source and target domains respectively, with a uniform prior $P(\theta_X = \theta_X^{\mathcal{S}}) = 0.5$, $P(\theta_X = \theta_X^{\mathcal{T}}) = 0.5$. Then, $I(\theta_X; (X, Y)) = JSD(P_{X,Y;\theta_X=\theta_X^{\mathcal{S}}} || P_{X,Y;\theta_X=\theta_X^{\mathcal{T}}})$, where JSD is the Jensen-Shannon Divergence, and $P_{X,Y;\theta_X=\theta_X^{\mathcal{S}}}$ and $P_{X,Y;\theta_X=\theta_X^{\mathcal{T}}}$ are the source and target domain joint distributions respectively.*

Therefore, to measure the influence of $\theta_X$ on the generating process as mimicked by our decoder, we need to measure the JSD between the joint distributions as implied by the reconstructed data, given by $\text{JSD}(P_{X,Y;\theta_X=\hat{\theta}_X^{\mathcal{S}}} || P_{X,Y;\theta_X=\hat{\theta}_X^{\mathcal{T}}})$ (note that here we are using inferred domain-specific variables $\hat{\theta}_X^{\mathcal{S}}$ and $\hat{\theta}_X^{\mathcal{T}}$). In theory, given enough data and expressiveness of the decoder, we can assume that we can learn to reconstruct the data perfectly. However, due to lack of labeled data in the target domain, we can only reconstruct the unlabeled features in the target domain, and have no access to $P_{X,Y;\theta_X=\hat{\theta}_X^{\mathcal{T}}}$.

To cope with this, we can use the inferred domain-specific variable $\hat{\theta}_X^{\mathcal{T}}$ to translate the source domain data to the target domain via the decoder: $\tilde{\mathbf{x}}_{trans.}^{\mathcal{S}} = \tilde{\phi}(\mathbf{z}^{\mathcal{S}}, \hat{\theta}_X^{\mathcal{T}})$. This translated data from the source to the target domain is labeled and can serve as an approximation of the joint distribution in the target domain, which we can use to constrain the JSD. A very intuitive way to minimize the JSD of the joint distributions of the source domain reconstructed and translated data, is to first define new random variables $V := [\tilde{X}^{\mathcal{S}}, Y^{\mathcal{S}}]^T$ and $V_{trans.} := [\tilde{X}_{trans.}^{\mathcal{S}}, Y^{\mathcal{S}}]^T$ as concatenations of the features and the label. Then, the JSD can be minimized by ensuring that the distributions of this joint feature-label vector is the same between the source reconstructed and translated data: $P(V) = P(V_{trans.})$. This can be achieved in several different ways, on of which is training an adversarial classifier on $V$ and $V_{trans.}$. However, since $Y$ is usually a scalar variable with a specific semantic meaning and often has a simple parametric form, concatenating it to $X$ would discard this prior knowledge, rendering this approach statistically inefficient.

Fortunately, the following theorem, adopted from [15], gives an upper bound of the JSD which is easy to minimize, and which makes use of the specific parametric form of $Y$ in the case of classification:

**Theorem 1 [15]:** *Let $P_{YX|\theta_X=\theta^{\mathcal{S}}}$ and $P_{YX|\theta_X=\theta^{\mathcal{T}}}$ denote the source and target domain distributions respectively. Let $Q_{Y|X}^{h_c}$ denote the conditional distribution of $Y$ given $X$ specified by the auxiliary classifier $h_c$. We have:*

$$JSD(P_{XY;\theta_X=\theta_X^{\mathcal{S}}} || P_{XY;\theta_X=\theta_X^{\mathcal{T}}}) \leq 2c_1 \sqrt{JSD(P_{X|\theta_X=\theta_X^{\mathcal{S}}} || P_{X|\theta_X=\theta_X^{\mathcal{T}}})}$$
$$+ c_2 \sqrt{KL(P_{Y|X,\theta_X=\theta_X^{\mathcal{S}}} || Q_{Y|X}^{h_c})} + c_2 \sqrt{KL(P_{Y|X,\theta_X=\theta_X^{\mathcal{T}}} || Q_{Y|X}^{h_c})}$$

,

where $c_1$ and $c_2$ are upper bounds of $\frac{1}{2} \int |P_{Y|X}(y|x)| \mu(x, y)$ and $\frac{1}{2} \int |Q_X(x)| \mu(x)$ respectively ($\mu$ is a $\sigma$-finite measure). Here, $\text{JSD}(P_{X|\theta_X=\hat{\theta}_X^{\mathcal{S}}} || P_{X|\theta_X=\hat{\theta}_X^{\mathcal{T}}})$ is equal to the JSD between the source and target domain marginal distributions of $X$ and is fixed, we can ignore this term when we minimize the upper bound using our algorithm. Therefore, to minimize the two KL divergences on the right-hand side:

$$\text{KL}(P_{Y|X,\hat{\theta}_X=\theta_X^{\mathcal{S}}} || Q_{Y|X}^{h_c}) \text{ and } \text{KL}(P_{Y|X,\theta_X=\hat{\theta}_X^{\mathcal{T}}} || Q_{Y|X}^{h_c}),$$

it suffices to train a joint auxiliary classifier $h_c$ on the reconstructed source domain data $\tilde{x}^{\mathcal{S}}$ and the translated data $\tilde{x}_{trans.}^{\mathcal{T}}$, whose task is to learn to predict the source domain label. Therefore, we introduce a cross-entropy loss $\mathcal{L}_{inv.}^{trans.}$ for the auxiliary classifier, whose two terms can be used to minimize the two KL divergences respectively:

$$\mathcal{L}_{inv.}^{trans} = -\sum_{j=1}^{C} \sum_{i=1}^{m_S} y_{ij} \log(\tilde{y}_{ij}^c) - \sum_{j=1}^{C} \sum_{i=1}^{m_S} y_{ij} \log(\tilde{y}_{ij,trans.}^c), \text{ where } \tilde{y}_i^c = h_c(\tilde{x}_i^{\mathcal{S}}),$$

and $\tilde{y}_{i,trans.}^c = h_c(\tilde{x}_{i,trans.}^{\mathcal{S}})$ are the softmax predictions of the reconstruction and the translation of the source domain respectively, using predictor $h_c$. Therefore, the total loss function can be formulated in the following manner:

$$\mathcal{L}_{total} = \lambda_\alpha \mathcal{L}_{reconst.} + \lambda_\delta \mathcal{L}_{classif.} - \lambda_\gamma \mathcal{L}_{inv.} + \lambda_\tau \mathcal{L}_{inv.}^{trans.} + \lambda_\kappa \mathcal{L}_{cent.},$$

and can be minimized by alternating optimization:

$$\min_{\phi,\tilde{\phi},h,h_c} \mathcal{L}_{total}$$
$$\min_{h_a} \lambda_\gamma \mathcal{L}_{inv.}.$$

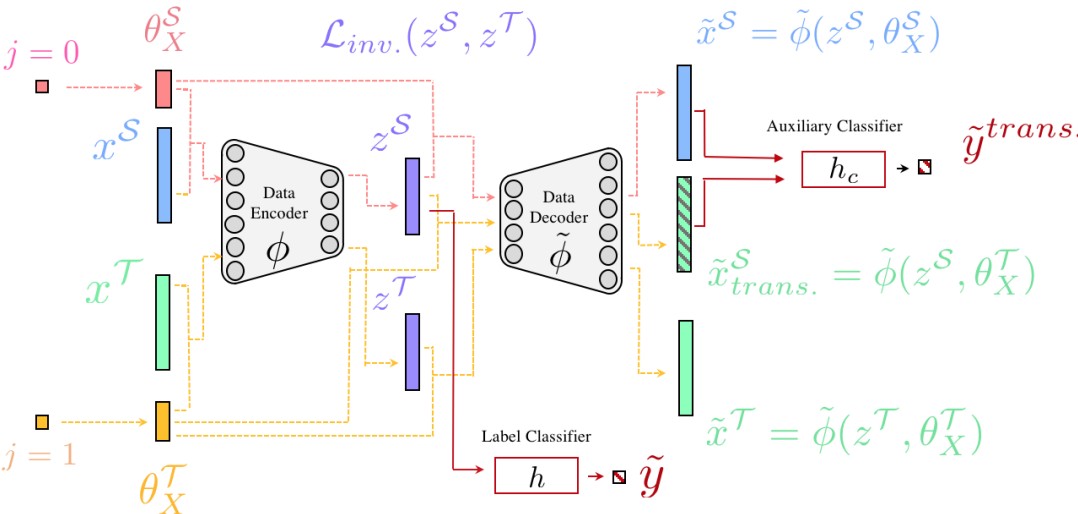

Figure 3: A diagram of the proposed autoencoder framework. Here, the domain index is mapped to $\theta_X$. Subsequently, the input data $\mathbf{x}$ and $\theta_X$ are mapped via $\phi$ to a latent representation $\mathbf{z}$, which in turn is reconstructed $\tilde{\mathbf{x}}$ using the decoder $\tilde{\phi}$, and $\theta_X$ as additional input. In addition, the penalty $\mathcal{L}_{inv.}$ is enforced to ensure invariance between $Z^{\mathcal{S}}$ and $Z^{\mathcal{T}}$. The hidden representation is also used to predict $\tilde{\mathbf{y}}$ in the source domain. Finally, $\mathcal{L}_{inv.}^{trans-y}$ enforces the minimal change of $P(X|Y)$ across domains. The pink and orange lines depict the flow of the source and target data and domain-specific information respectively.

This alternating minimization problem can be solved using Gradient-Reversal Layers (GRL), as shown in [13]. $\mathcal{L}_{inv.}$ can also be expressed in terms of Maximum Mean Discrepancy (MMD) [16]. Our model is now equipped with a way to ensure that when reconstructing the data, $\theta_X$ cannot play a major role in the decoder $\tilde{\phi}$. Since $\theta_X$ correspond to minimal changes of the conditional distribution $P(X|Y)$, this forces $Z$ to retain $Y$-specific structure, and in turn implicitly regularizes the encoder $\phi$, resulting in a conditionally invariant representation. In the following section, we will empirically demonstrate this property of the constraint.

## 4   Empirical Evaluation

In order to provide intuition about the proposed method's advantages in the challenging scenarios described above, we first designed a simple but informative 2D simulated example for domain adaptation using mixtures of Gaussian distributions. This simulated dataset is presented on Figure 4(a), and is representative of the example discussed in Figure 2, since there is a significant region of overlap between the two domains, in which the points have opposite labels across domains (details about the dataset can be found in the supplementary materials). In this experiment, we compared our method DSAN, with the following baselines: (1) DSAN-unreg., which is our proposed method but without regularizing the transformation $\phi$ using the mutual information minimization in our loss; (2) DSN, the method proposed in [4], which makes use of domain-specific private encodings, but which are only used for reconstruction of the data, and not for inferring the shared invariant representation $Z$; (3) DANN, the classical domain-adversarial method proposed by [13]. We used a 2D hidden

representation $Z$ (more details about implementation and tuning can be found in the supplement).

The results are presented in Figure 4b, demonstrating that our approach greatly outperforms the baselines. To visualize how our algorithm is able to solve this challenging example with much greater success, we also present the 2D invariant representations in the source and target domains on Figure 6. At the top, we have the hidden representations $Z$ of DSAN-unreg., and on the bottom the ones yielded by our method DSAN (run on a separate random initialization). On the left side we present $Z$ in the source domain, labeled by the true values, in the middle we show this representation in the target domain labeled with the prediction of the algorithm, and on the right-hand side we present $Z$ in the target domain labeled with the true labels. From this figure, one can appreciate that the unregularized version of our approach yields a representation $Z$ which discards a lot of information about $Y$, as can be seen in parts 5b and 5c. On the other hand, parts 5e and 5f show that our algorithm with the mutual information minimization constraint can successfully preserve label-specific information in the invariant representation.

### 4.1 Real Data Experiments

To evaluate our method on real datasets, we consider three datasets and respective tasks from various domains of applications: cross-domain Wi-Fi localization, Amazon product reviews and image classification. For detailed descriptions of the experimental design, hyperparameter tuning and neural network architectures, as well as ablation studies, we refer the interested reader to the supplementary materials. We performed evaluation on the following datasets: **Wi-Fi localization**: introduced in [49], is a dataset in which wireless signal data was collected in a hallway area discretized as a grid. In each grid, data was recorded from 67 access points (dimensions). The task is to predict the location from the signals (which has been converted to a classification problem with 19 classes). There are three domains in this dataset (with 1140 points each), collected in three different time points. This is a low-dimensional dataset, in which we verified via manual inspection that both scenarios of support overlap described in Section 2 hold; **Amazon Review:** Amazon Review is another benchmark for multi-domain sentiment analysis. It contains positive and negative reviews of four kinds of products: Kitchen appliance (K), DVDs (D), Electronics (E), and Books (B). **ImageCLEF:** A standard UDA benchmark dataset for image classification, consisting of three domains: *Caltech-256*(C), *ImageNet ILSVRC*(I), and *Pascal VOC2012*(P), consisting of 12 classes. **Baselines:** We compare DSAN with classical approaches like TCA [28] and GFK [14], as well as with some deep transfer learning models. Three recently proposed methods, DSR [5], DIRT-T [32], MDD [51], BSP [7], MSTN [42] and RSDA [17] are included. For both the real-world and simulated experiments, we ran and tuned the baseline methods ourselves (all baselines for Wi-Fi and simulated, and MDD and DIRT-T for Amazon dataset).

The classification accuracies on the Amazon-Review Dataset for unsupervised domain adaptation are shown in Table 2, which shows that our algorithm model outperforms the baselines in the vast majority of source-target directions. We also perform a Wilcoxon signed-rank test on the experiment result of different random seed, and the p-value is 0.0216. The performance of our method on the Wi-Fi localization dataset is shown on Table 1. In this experiment, we sub-sampled 950 points in each domain to create 10 replicate experiments, and we present the accuracies and standard deviations across all replicates. From the results, one can appreciate that our method outperforms all baselines, for the majority of the pairs. We calculated a Wilcoxon p value $p = 0.002$ across all replicate experiments combined. The performance of our method on the ImageCLEF dataset is presented on Table 3, in which one can appreciate that we achieve state-of-the-art performance on most transfer directions. For this experiment, certain baselines such as MDD and RSDA use a pre-trained ResNet neural network architecture as the transformation function $\phi$. For the proposed method, we instead used features obtained from a pre-trained ResNet architecture, and we incorporated them into the autoencoder neural network described in the previous section (more details on the architecture and hyperparameters can be found in the supplementary information). While we currently obtain state-of-the-art performance on this dataset, we believe that adapting our method to make use of ResNet fine-tuning will further increase the performance.

## 5 Conclusion

In this paper, we have demonstrated that the existing paradigm of representation learning for domain adaptation using a fixed encoder $\phi$ of the data, can be sub-optimal when the same points can

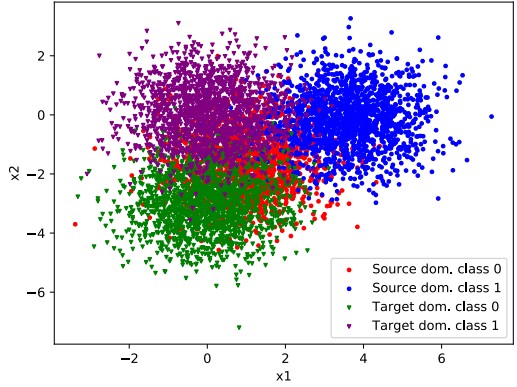

| Model | Accuracy |
|---|---|
| DANN | 58.2±4.3 |
| DSN | 58.8±5.2 |
| DSAN-unreg. | 79.4±2.2 |
| DSAN | **87**±1.0 |

(b) Results on the simulated data.

(a) Scatter plot of simulated data

Figure 4

Table 1: Accuracy (%) on Wi-Fi localization dataset for unsupervised domain adaptation

| Models | t1 → t2 | t1 → t3 | t2 → t1 | t2 → t3 | t3 → t1 | t3 → t2 | Average |
|---|---|---|---|---|---|---|---|
| DANN [12] | 31.30±1.3 | 33.15±1.9 | 38.03±2.0 | 32.41±2.3 | 28.54±0.8 | 31.12±1.6 | 32.59 |
| DSN [4] | 31.25±1.5 | 34.23±2.4 | 34.32±2.8 | 29.47±0.7 | 27.0±1.6 | 31.8±1.3 | 31.35 |
| DSR [5] | 34.20±1.3 | 34.62±4.8 | 30.61±3.5 | 34.72±1.1 | 33.13±0.5 | 31.61±0.9 | 31.13 |
| MDD [51] | 30.92±2.1 | 34.42±1.8 | 30.01±1.5 | 27.58±1.8 | 33.25±1.0 | 27.34±2.3 | 30.59 |
| DIRT-T [32] | 36.38±1.2 | 33.66±2.7 | 37.12±3.4 | 31.85±2.87 | 33.98±1.42 | 28.48±2.68 | 33.58 |
| BSP [7] | 35.5±1.3 | 32.16±2.1 | 31.15±1.4 | 31.06±1.9 | 32.91±1.1 | 33.83±2.2 | 32.72 |
| DSAN-U | 31.16±1.1 | 29.01±1.7 | 32.76±0.9 | 32.18±1.1 | 31.02±1.7 | **33.62**±**1.8** | 31.45 |
| DSAN | **40.45**±**1.6** | **37.35**±**2.3** | **38.22**±**2.4** | **39.94**±**2.1** | **36.47**±**3.5** | 34.18±2.9 | **37.77** |

have different labels across domains. We have introduced a method which takes domain-specific information in order to provide the data encoder with the necessary flexibility and appropriate constraints, in order to retain valuable information about $Y$ in the latent representation $Z$. We note that in many real-world applications with high-dimensional datasets, the supports of the source and target domains may not overlap. In this case, domain-specific information is contained in the features $X$ and can be automatically used by the encoder $\phi(X)$. However, $\phi(X)$ would then have the same problem with excessive flexibility and can still suffer from finding trivial representations of $Z$ that discard information about $Y$. Therefore, our principle for constraining domain-specific encoders would still be beneficial in this scenario. For future work, a promising direction is combining this framework motivated by the data-generating process with more powerful encoders, such as the spherical neural network proposed by [17].

# 6 Acknowledgements

We are very grateful to the anonymous reviewers for their help in improving the paper. This work was supported in part by the National Institutes of Health (NIH) under Contract R01HL159805, by the NSF-Convergence Accelerator Track-D award #2134901, by the United States Air Force under Contract No. FA8650-17-C7715, and by a grant from Apple Inc. GM was supported by Australian Research Council Project DE210101624. Ruichu and Zijian were supported in part by National Science Fund for Excellent Young Scholars (62122022), Natural Science Foundation of China (61876043, 61976052).

Table 2: Accuracy (%) on Amazon Review dataset for unsupervised domain adaptation

| Methods | B→D | B→E | B→K | D→B | D→E | D→K | E→D | E→B | E→K | K→B | K→D | K→E | Avg |
|---|---|---|---|---|---|---|---|---|---|---|---|---|---|
| NN | 49.6 | 49.8 | 50.3 | 53.3 | 51.0 | 53.1 | 50.8 | 50.9 | 51.2 | 52.2 | 51.2 | 52.3 | 51.3 |
| TCA [28] | 63.6 | 60.9 | 64.2 | 63.3 | 64.2 | 69.1 | 59.5 | 62.1 | 74.8 | 64.1 | 65.4 | 74.5 | 65.5 |
| GFK [14] | 66.4 | 65.5 | 69.2 | 66.3 | 63.7 | 67.7 | 62.4 | 63.4 | 73.8 | 65.5 | 65.0 | 73.0 | 66.8 |
| SA [11] | 67.0 | 70.8 | 72.2 | 67.5 | 67.1 | 69.4 | 61.4 | 64.9 | 70.4 | 64.4 | 64.6 | 68.2 | 67.3 |
| BDA [39] | 64.2 | 62.1 | 65.4 | 62.4 | 66.3 | 68.9 | 59.6 | 61.6 | 74.7 | 62.7 | 64.3 | 74.0 | 65.5 |
| CORAL [36] | 71.6 | 65.1 | 67.3 | 70.1 | 65.6 | 67.1 | 67.1 | 66.2 | 77.6 | 68.2 | 68.9 | 75.4 | 69.1 |
| JGSA [45] | 66.6 | 75.0 | 72.1 | 55.5 | 67.3 | 65.6 | 51.6 | 50.8 | 55.0 | 58.3 | 56.4 | 51.7 | 60.5 |
| DANN [12] | 78.4 | 73.3 | 77.9 | 72.3 | 75.4 | 78.3 | 71.3 | 73.8 | 85.4 | 70.9 | 74.0 | 84.3 | 64.9 |
| MDD [51] | 77.1 | 74.4 | 77.0 | 74.7 | 74.1 | 76.3 | 72.4 | 70.2 | 83.3 | 69.3 | 73.2 | 82.8 | 75.4 |
| DIRT-T [32] | 78.6 | 76.1 | 75.5 | 76.8 | 75.2 | 79.1 | 69.6 | 71.0 | 84.2 | 69.2 | 73.3 | 79.5 | 75.7 |
| EasyTL [40] | 79.8 | 79.7 | 80.9 | 79.9 | 80.8 | 82.0 | 75.0 | **75.3** | 84.9 | **76.5** | 76.3 | 82.5 | 79.5 |
| BSP [7] | 79.33 | 73.86 | 75.86 | 75.67 | 74.77 | 77.2 | 72.81 | 71.3 | 84.02 | 70.92 | 73.59 | 84.29 | 76.14 |
| RSDA [17] | 80.1 | 77.8 | 84.4 | 76.6 | 79.4 | 82.3 | 73.6 | 71.1 | 86.6 | 71.8 | 74.7 | 84.4 | 77.9 |
| DSAN-U | 81.0 | 78.6 | 78.9 | 78.4 | 79.4 | 83.3 | 76.2 | 74.5 | 87.4 | 73.1 | 77.0 | 83.0 | 78.9 |
| DSAN | **82.7** | **80.8** | **82.6** | 79.5 | **81.4** | **85.3** | **76.7** | 75.1 | **88.0** | 73.8 | **77.3** | **85.0** | **80.7** |

Table 3: Accuracy (%) on ImageCLEF dataset for unsupervised domain adaptation (* reproduced by [17])

| Models | I → P | P → I | I → C | C → I | C → P | P → C | Average |
|---|---|---|---|---|---|---|---|
| DResNet-50 [18] | 74.8± 0.3 | 83.9± 0.1 | 91.5 ± 0.3 | 78.0 ± 0.2 | 65.5 ±0.3 | 91.2 ± 0.3 | 80.7 |
| iCAN [50] | 79.5 | 89.7 | 94.7 | 89.9 | 78.5 | 92.0 | 87.4 |
| CDAN [23] | 77.7 ± 0.3 | 90.7±0.2 | 97.±0.3 | 91.3±0.3 | 74.2±0.2 | 94.3±0.3 | 87.7 |
| SymNets [52] | 80.2±0.3 | 93.6±0.2 | 97.0±0.3 | 93.4±0.3 | 78.7±0.3 | 96.4±0.1 | 89.9 |
| SAFN+ENT [43] | 79.3±0.1 | 93.3±0.4 | 96.3±0.4 | 91.7±0.0 | 77.6±0.1 | 95.3±0.1 | 88.9 |
| CAT [10] | 77.2±0.2 | 91.6±0.3 | 95.5±0.3 | 91.3±0.3 | 75.3±0.6 | 93.6±0.5 | 87.3 |
| DANN [12] | 75.0±0.6 | 86.0±0.3 | 96.2±0.4 | 87.0±0.5 | 74.3±0.5 | 91.5±0.6 | 85.0 |
| DANN+S [17] | 78.3±0.5 | 91.0±0.4 | 96.8±0.2 | 91.8±0.6 | 77.7±0.5 | 95.2±0.5± | 88.5 |
| RSDA-DANN [17] | 79.2±0.4 | 93.0±0.2 | **98.3**±0.4 | 93.6±0.4 | 78.5±0.3 | **98.2**±0.2 | 90.1 |
| RSDA-MSTN [17] | 79.8±0.2 | 94.5±0.5 | 98.0±0.4 | 94.2±0.4 | 79.2±0.3 | 97.3±0.3 | 90.5 |
| MSTN* [42] | 77.3±0.3 | 91.3±0.4 | 96.8±0.2 | 91.2±0.5 | 77.7±0.2 | 95.0±0.5 | 88.2 |
| DSAN-U | 79.2±0.2 | 94.1±0.1 | 96.2±0.2 | 93.5±0.5 | 77.4±0.8 | 90.0±0.3 | 88.3 |
| DSAN | **80.3**±0.5 | **95.2**±0.3 | 97.5±0.2 | **94.7**±0.5 | **79.6**±0.6 | 96.7±0.2 | **90.7** |

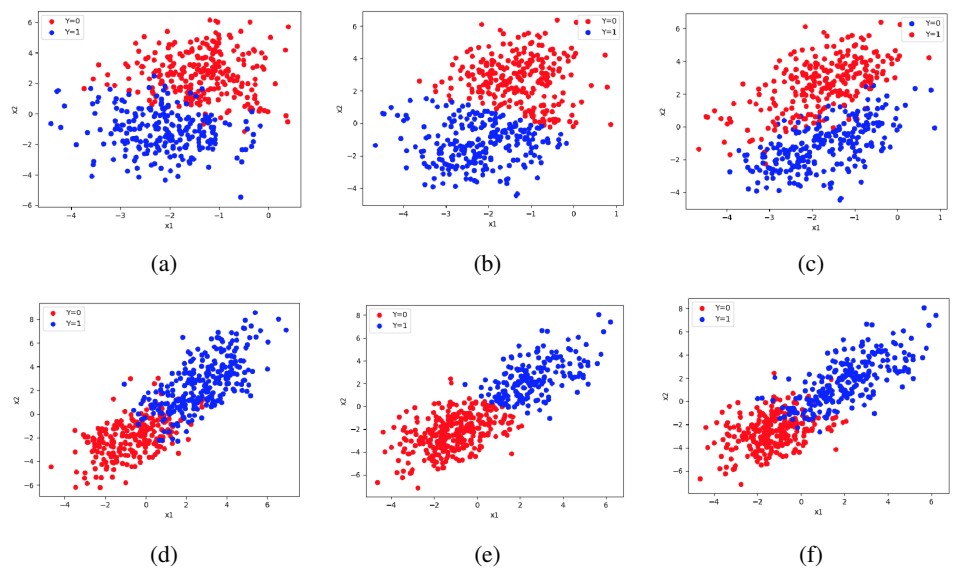

(a)    (b)    (c)

(d)    (e)    (f)

Figure 5:
$Z$ in the simulations. First three scatter-plots show it without regularization: (a) $Z$ of the source domain labeled with true labels, (b) $Z$ of the target domain labeled with the predictions (c) $Z$ of the target domain with the true labels. (d), (e), and (f) show the same, but with regularization.

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
