# Supplementary Materials For: "Domain Adaptation with Invariant Representation Learning: What Transformations to Learn?"

**Petar Stojanov**[1,4*], **Zijian Li**[2*], **Mingming Gong**[3],
**Ruichu Cai**[2], **Jaime G. Carbonell**[1], **Kun Zhang**[1]
[1] Carnegie Mellon University
[2] School of Computer Science, Guangdong University of Technology
[3] School of Mathematics and Statistics, University of Melbourne
[4] Broad Institute of MIT and Harvard

## 1 Proofs of Theoretical Statements

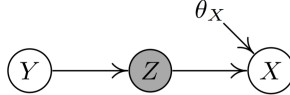

Figure 1: The underlying data-generating process under conditional shift, of the observed variables $Y$ and $X$, and the latent variable $Z$. $\theta_X$ represents the changing parameters of $P(X|Y)$ across domains.

In this section we provide proofs of theoretical statements in the paper.

### 1.1 Proof of Proposition 1:

Before proving this proposition, let us re-state it:

**Proposition 1:** *Let $A \subset \mathcal{X}$ be a region in input space such that $P^{\mathcal{S}}(X \in A) > 0$ and $P^{\mathcal{T}}(X \in A) > 0$, and $P^{\mathcal{S}}(X \in A) \neq P^{\mathcal{T}}(X \in A)$. Furthermore, let $\phi : \mathcal{X} \to Z$ be an encoder s.t. $A = \{a : \phi(a) \in B\}$ for some $B \subset \mathcal{Z}$. Then, there is no function $\phi$ s.t. $P^{\mathcal{S}}(\phi(X) \in B) = P^{\mathcal{T}}(\phi(X) \in B)$.*

*Proof.* Let $\mathcal{X}$ be the input feature space, and $\mathcal{Z}$ be the latent space in which the invariant representation is learned via encoder $\phi : \mathcal{X} \to \mathcal{Z}$.

Let there be a subset in the invariant space $B \subset \mathcal{Z}$, and suppose that we have marginal invariance in the latent space: $P^{\mathcal{S}}(\phi(X) \in B) = P^{\mathcal{T}}(\phi(X) \in B), \forall B$. Define the pre-image of $B$ as:

$$A = \{a \in \mathcal{X} : \phi(a) \in B\}.$$

Then, we have the following equality relationships:

$$P^{\mathcal{S}}(\phi(X) \in B) = P^{\mathcal{S}}(X \in A) = P^{\mathcal{T}}(\phi(X) \in B) = P^{\mathcal{T}}(X \in A),$$

which implies that $P^{\mathcal{S}}(X \in A) = P^{\mathcal{T}}(X \in A)$ must hold. This further implies that if there is difference in the source and target distributions in the support-overlapping region $A$, given by:

$$P^{\mathcal{S}}(X \in A) \neq P^{\mathcal{T}}(X \in A),$$

---

*These authors contributed equally to this work.

35th Conference on Neural Information Processing Systems (NeurIPS 2021), Sydney, Australia.

then the following must hold: $P^{\mathcal{S}}(\phi(X) \in B) \neq P^{\mathcal{T}}(\phi(X) \in B)$. Therefore, if there is a region of support overlap for which the distributions of the source and target domain are different, a single transformation $\phi$ cannot be used to map data in this region to a marginally-invariant representation. $\qquad\square$

## 1.2 Proof of Proposition 2:

Before proving this proposition, let us re-state it:

**Proposition 2:** *Let the true labeling functions in the source and target domain be $f_S, f_T : \mathcal{X} \to \mathcal{Y}$, respectively. Let $\mathcal{A} \subseteq \mathcal{X}$ be a region s.t. $f_S(a) \neq f_T(a), \forall a \in \mathcal{A}$. Let $g : \mathcal{X} \to \mathcal{Y}$ be a composition of a representation learner $\phi : \mathcal{X} \to \mathcal{Z}$ and a classifier $h : \mathcal{Z} \to \mathcal{Y}$. If $\phi$ is the same function across domains, then for a 0-1 loss, the risk over the region $\mathcal{A}$ is $\epsilon^{\mathcal{A}}(g) = \epsilon_S^{\mathcal{A}}(g) + \epsilon_T^{\mathcal{A}}(g) \geq 1$.*

*Proof:* The total risk for the 0-1 loss is:

$$\epsilon^{\mathcal{A}}(g) = \epsilon_S^{\mathcal{A}}(g) + \epsilon_T^{\mathcal{A}}(g) = \Pr(g(a) \neq f_S(a)) + \Pr(g(a) \neq f_T(a)). \tag{1}$$

Since $f_S(a) \neq f_T(a), \forall a \in \mathcal{A}$, $g$ either predicts the correct label for $a$ in one domain and not the other, or predicts the wrong label in both. Therefore, for the multiple-class classification problem, the following inequality holds:

$$\Pr(g(a) \neq f_T(a)) \geq 1 - \Pr(g(a) \neq f_S(a)), \tag{2}$$

so it follows that:

$$\epsilon^{\mathcal{A}}(g) = \epsilon_S^{\mathcal{A}}(g) + \epsilon_T^{\mathcal{A}}(g) = \Pr(g(a) \neq f_S(a)) + \Pr(g(a) \neq f_T(a)) \geq 1.$$

$\qquad\square$

## 1.3 Proof of Lemma 1:

Before proving this lemma, we shall first state it here:

**Lemma 1:** *Let $X, Y \sim P(X, Y; \theta_X)$, and let $\theta_X$ be a discrete random variable with possible values $\theta_X^S$ and $\theta_X^T$ in the source and target domains respectively, with a uniform prior $P(\theta_X = \theta_X^S) = 0.5$, $P(\theta_X = \theta_X^T) = 0.5$. Then, $I(\theta_X; (X, Y)) = JSD(P_{X,Y;\theta_X=\theta^S} || P_{X,Y;\theta_X=\theta^T})$, where JSD is the Jensen-Shannon Divergence.*

*Proof:* From the general definition of mutual information, and the by factoring the joint distribution according to the generating process in Figure 1 we have:

$$I(\theta_X; (X, Y)) = \int p_{XY\theta_X}(x, y, \theta) \log \frac{p_{XY\theta_X}(x, y, \theta)}{p_{\theta_X}(\theta) p_{XY}(x, y)} dx dy d\theta$$

$$= \int p_{XY\theta_X}(x, y, \theta) \log \frac{p_{XY|\theta_X}(x, y|\theta)}{p_{XY}(x, y)} dx dy d\theta.$$

Since $\theta_X$ is a discrete random variable with a uniform prior, the above expression can be rewritten as:

$$I(\theta_X; (X, Y)) = \frac{1}{2} \int p_{XY|\theta_X=\theta_X^S}(x, y) \log \frac{p_{XY|\theta_X=\theta_X^S}(x, y)}{p_{XY}(x, y)} dx dy$$

$$+ \frac{1}{2} \int p_{XY|\theta_X=\theta_X^T}(x, y) \log \frac{p_{XY|\theta_X=\theta_X^T}(x, y)}{p_{XY}(x, y)} dx dy$$

$$= \frac{1}{2} \mathrm{KL}(P_{XY|\theta_X=\theta_X^S} || P_{XY}) + \frac{1}{2} \mathrm{KL}(P_{XY|\theta_X=\theta_X^T} || P_{XY})$$

$$= \mathrm{JSD}(P_{XY|\theta_X=\theta_X^S} || P_{XY|\theta_X=\theta_X^T}). \tag{3}$$

$\qquad\square$

## 2 Enforcing Invariance with Maximum Mean Discrepancy

When the input dataset is low-dimensional, it is also possible to enforce marginal invariance for the representation $Z$ using MMD (Maximum Mean Discrepancy) [4] between the representations $Z$ in the source and target domains, instead of adversarial training. Let $\mathbf{z}_i^{\mathcal{S}}, \mathbf{z}_i^{\mathcal{T}}$ denote the $i$-th instance of $Z$ in the source and target domains respectively. Then, the MMD loss can be written as follows:

$$\mathcal{L}_{inv.}^{MMD} = \frac{1}{m_S^2} \sum_{i,j=1}^{m_S} \kappa(\mathbf{z}_i^{\mathcal{S}}, \mathbf{z}_j^{\mathcal{S}}) + \frac{1}{m_S m_T} \sum_{i,j=1}^{m_S, m_T} \kappa(\mathbf{z}_i^{\mathcal{S}}, \mathbf{z}_j^{\mathcal{T}}) + \frac{1}{m_T^2} \sum_{i,j=1}^{m_T} \kappa(\mathbf{z}_i^{\mathcal{T}}, \mathbf{z}_j^{\mathcal{T}}).$$

We followed the procedure in [2], and used a mixture kernel function of $q$ RBF kernels: $\kappa(\mathbf{z}_1, \mathbf{z}_2) = \sum_{i=1}^{q} \eta_i \exp\{-||\mathbf{z}_1 - \mathbf{z}_2||^2\}/\sigma_i^2$, where $\sigma_i^2$ is the kernel width of the $i$-th kernel, and $\eta_i$ is a mixing weight which we set to $1/q$. For all experiments where we used MMD, we used the following RBF kernels (indexed by the respective kernel width): $\sigma_i^2 = [10^{-2}, 10^{-1}, 1, 5, 10, 50, 100, 500, 1000, 5000, 10000]$. Then, the total loss can be modified as follows:

$$\mathcal{L}_{total} = \lambda_\alpha \mathcal{L}_{reconst.} + \lambda_\delta \mathcal{L}_{classif.} + \lambda_\gamma \mathcal{L}_{inv.}^{MMD} + \lambda_\tau \mathcal{L}_{inv.}^{trans.}, \tag{4}$$

and in this case, alternating optimization is not needed. Then, we can simply minimize this total loss:

$$\min_{\phi, \tilde{\phi}, h, h_c} \mathcal{L}_{total}. \tag{5}$$

## 3 Using Pseudo-Labels and Ablation Study

In the main text, for the ImageCLEF dataset, in addition to $\mathcal{L}_{total}$ as described in the methods section, we make use of pseudo-labels to refine the quality of the invariant latent representation $Z$. For this purpose, we make use of the centroid procedure described in [10]. Namely, for this dataset we introduce an additional term to the loss. Given $C$ class labels and a distance metric $\Phi$ (we use Euclidean distance), we have:

$$\mathcal{L}_{SM} = \sum_{j=1}^{C} \Phi(C_j^{\mathcal{S}}, C_j^{\mathcal{T}}),$$

where $C_j^{\mathcal{S}}, C_j^{\mathcal{T}}$ are the centroids for the $j$-the class in the source and target domains respectively. Thus, we add this loss function to our total loss, in addition to the other terms described in the main text.

### 3.1 Ablation Study

To demonstrate the efficacy of the proposed procedure, we performed ablation studies to assess the contribution of the following design choices:

- the various terms in our total loss function,
- the need to have domain-specific information in the encoder and decoder

In Table 1 we present the results. Here, the columns are described as follows:

- "Dec." indicates whether a decoder is being used at all.
- "$\theta$-Enc." indicates whether domain-specific information is being used in the encoder.
- "$\theta$-Dec." indicates whether domain-specific information is being used in the decoder.
- "Entropy" indicates whether the conditional-entropy loss is being used.
- "Pseudo-labels" indicates whether pseudo-labels are being used.

From the results, a simple conclusion that can be derived is that the constraining the decoder according to the minimality of the generating process (via mutual information minimization) is instrumental to the performance of our method, and to attaining state-of-the-art performance.

| Dec. | $\theta$-Enc. | $\theta$-Dec. | Entropy | Pseudo | Reg. | I→P | P→I | I→C | C→I | C→P | P→C | Avg. |
|---|---|---|---|---|---|---|---|---|---|---|---|---|
| ✗ | ✓ | N/A | ✓ | ✓ | ✓ | 78.00±0.9 | 92.40±2.8 | 95.7±0.6 | 92.77±0.9 | 75.3±1.4 | 92.77±0.4 | 87.77 |
| ✓ | ✓ | ✗ | ✓ | ✓ | ✓ | 79.15±0.2 | 92.96±0.6 | 96.44±0.1 | 93.61±0.5 | 77.22±0.1 | 92.44±2.9 | 88.63 |
| ✓ | ✗ | ✓ | ✓ | ✓ | ✓ | 79.01±0.5 | 93.22±0.6 | 96.33±0.2 | 93.28±0.4 | 77.33±2.7 | 94.49±0.7 | 89.58 |
| ✓ | ✓ | ✓ | ✗ | ✗ | ✓ | 79.78±0.2 | 91.59±2.2 | 94.61±0.3 | 93.72±0.2 | 73.90±2.9 | 94.71±0.9 | 88.05 |
| ✓ | ✓ | ✓ | ✗ | ✓ | ✓ | 79.39±0.5 | 93.01±2 | 95.89±0.8 | 93.50±0.5 | 77.94±1.2 | 94.11±1.2 | 88.97 |
| ✓ | ✓ | ✓ | ✗ | ✓ | ✓ | 79.61±0.3 | 94.61±0.3 | 95.89±0.5 | 93.61±0.4 | 77.99±0.4 | 95.79±0.8 | 89.58 |
| ✓ | ✓ | ✓ | ✓ | ✓ | ✗ | 79.22±0.2 | 94.11±0.1 | 96.17±0.2 | 93.5±0.5 | 77.44±0.8 | 92.0±2.3 | 88.74 |
| ✓ | ✓ | ✓ | ✓ | ✓ | ✓ | **80.33**±0.5 | **95.23**±0.3 | **97.54**±0.2 | **94.66**±0.5 | **79.61**±0.6 | **96.67**±0.2 | **90.7** |

Table 1: Ablation study on the ImageCLEF dataset.

# 4 Experimental Setting, Neural Network Architectures, and Hyperparameters

In this section we provide details regarding the empirical evaluation in the main text. We note that unless mentioned otherwise, all non-linearity activations of hidden units in the neural network model are ReLU. We also note that in all of our experiments $\theta_X$ is one-dimensional and parameterized by a linear function of $j$, the domain index. Furthermore, in our implementation, the inputs $X$ and $Z$ were concatenated with $\theta_X$ before being passed to the encoder and decoder $\phi$ and $\tilde{\phi}$ respectively.

## 4.1 Simulated Experiments

The simulated dataset was generated in the following manner: we first let $Z \sim 0.5\mathcal{N}([0,0]^T, \mathbf{I}_2) + 0.5\mathcal{N}([0,3]^T, \mathbf{I}_2)$, where the two mixtures correspond to labels $Y = 0$ and $Y = 1$ respectively, where $P(Y = 0) = P(Y = 1) = 0.5$, and $\mathbf{I_2}$ corresponds to a $2 \times 2$ identity matrix. Then, we let the source domain data $X^S = \mathbf{R}(Z + [2,0]^T)$ and $X^T = Z + [0,-3]^T$, where $\mathbf{R}$ is a rotation matrix which rotates the data 45 degrees clockwise.

We performed 10 replicate experiments, in which we generated 3200 points in the source and target domains each, as described above. For each replicate, the same random seed was used for each method. As mentioned in the main text, all baselines and the proposed method used the same architecture:

- **The encoder** $\phi$ is a multi-layer perceptron (MLP) with two hidden layers of 128 dimensions each.

- **The invariant representation** $Z$ has 2 dimensions, so that it can be easily visualized.

- **The changing parameters** $\theta_X$ have 1 dimension and are a linear function of the domain index.

- **The classifier** $h$ is an MLP of a single hidden layer of 128 dimensions.

- **The decoder** $\tilde{\phi}$ (for DSAN and DSAN-unreg.) is an MLP with two hidden layers of 128 dimensions each.

- **The regularization (auxiliary) classifier:** $h_c$ is an MLP with with two hidden layers of 512 dimensions each.

- **The adversarial classifier :** $h_a$ has two hidden layers of 32 dimensions each, although we were also able to use MMD for this setting and forego $h_a$ altogether.

The hyper-parameters used for training are presented on Table 2.

## 4.2 Wireless Localization Experiments

Similarly to the simulated experiment, we performed 10 replicate experiments subsampling 950 points from a total of 1140, for each of the three domains. Furthermore, we used the same architecture of the neural network for the baselines and for the proposed method, where applicable. The neural network architecture can be summarized as follows:

- **The encoder** $\phi$ is a multi-layer perceptron (MLP) with two hidden layers of 512 dimensions each.

| Hyperparameter: | Value: |
|---|---|
| learning rate | 0.011 |
| $\lambda_\alpha$ | 0.1 |
| $\lambda_\gamma$ | 0.2 |
| $\lambda_\delta$ | 0.5 |
| batch size | 128 |
| optimizer | SGD |
| $L_2$ weight decay | 1.00E-06 |
| dropout | 0 |
| batch normalization | 0 |
| $\lambda_\tau$ | 1 |

Table 2: Hyperparameter settings for the simulated dataset experiments.

- **The invariant representation** $Z$ is of 20 dimensions.
- **The changing parameters** $\theta_X$ have 1 dimension and are a linear function of the domain index.
- **The classifier** $h$ is an MLP of a single hidden layer of 128 dimensions with ReLU activation.
- **The decoder** $\tilde{\phi}$ (for DSAN, DSAN-unreg., and DSN) is an MLP with two hidden layers of 512 dimensions each.
- **The regularization (auxiliary) classifier:** $h_c$ is an MLP with with two hidden layers of 512 dimensions each.

For this experiment, to enforce invariance we used the MMD formulation of our approach instead of adversarial training. Each replicate experiment was run for 400 epochs, and the accuracies reported are the ones at the last epoch. The hyperparameter settings can be found on Table 3.

| Hyperparameter: | Value: |
|---|---|
| learning rate | 0.001 |
| $\lambda_\alpha$ | 0.01 |
| $\lambda_\gamma$ | 0.4 |
| $\lambda_\delta$ | 0.4 |
| batch size | 128 |
| optimizer | SGD |
| $L_2$ weight decay | 1.00E-06 |
| dropout | 0 |
| batch normalization | 0 |
| $\lambda_\tau$ | 2 |

Table 3: Hyperparameter settings for the wireless localization dataset experiments.

We also considered MDD [1] and DIRT-T [7] on this dataset. For each baseline, the encoder was selected among MLP dimensionalities 64, 128 and 512, and between one and two layers in depth. The same dimensionalities were used as options for the hidden representation $Z$. We selected the one with best performance in each respective baseline. The hyperparameter settings used are as follows:

- for MDD, the encoder consists of two layers of 128 dimensions each. The hidden representation is 128, and the learning rate is 0.01. For this experiment, we used batch normalization
- for DIRT-T, the encoder consists of two layers of 64 dimensions each. The hidden representation is 64, and the learning rate is 2e-03. For this experiment, we did not use batch normalization

### 4.3 Sentiment Prediction in the Amazon Product Reviews

For the experiments on the Amazon dataset, we performed 5 runs on different random seeds using the entire dataset. Each replicate experiment was run for 150 epochs, and the accuracies reported are the best ones across epochs. We provide the details regarding the architecture used below:

- **The encoder** $\phi$ is a multi-layer perceptron (MLP) with a single hidden layer of 4000 dimensions.

- **The invariant representation** $Z$ is of 4096 dimensions.

- **The changing parameters** $\theta_X$ have 1 dimension and are a linear function of the domain index.

- **The classifier** $h$ is an MLP of a single hidden layer with 100 dimensions.

- **The decoder** $\tilde{\phi}$ is an MLP with a single layer of 5000 dimensions (which is the input dimensionality of the Amazon Product Reviews data. It is only a linear transformation with ReLU activation, and no hidden layers).

- **The domain-adversarial discriminator** $h_a$ is an MLP of a single hidden layer with 100 dimensions.

- **The regularization (auxiliary) classifier:** $h_c$ is an MLP with two hidden layers of 3000 dimensions each.

The hyperparameter setting used for this experiment is presented on Table 6.

| Hyperparameter: | Value: |
|---|---|
| learning rate | 1.00E-04 |
| $\lambda_\alpha$ | 1 |
| $\lambda_\gamma$ | 1 |
| $\lambda_\delta$ | 1 |
| batch size | 128 |
| optimizer | Adam |
| $L_2$ weight decay | 1.00E-05 |
| dropout | 0.75 |
| batch normalization | 0 |
| $\lambda_\tau$ | 0.005 |

Table 4: Hyperparameter settings for the Amazon Product Reviews dataset experiment.

Regarding the baselines for this experiment, for a fair comparison, the result of most baselines are directly reported from their original papers wherever available. In order to include more recent state-of-the-art baselines, we also compare our model with DIRT [7] and MDD [1]. For these two baselines, we ran the experiments ourselves using the respective source codes. The hyper-parameter settings are shown in Table 5. To select the hyperparameters in Table 5, we try different hyperparameter

|  | RSDA | DIRT-T | MDD |
|---|---|---|---|
| dropout rate | 0.5 | 0.5 | 0.5 |
| feature demension | 4096 | 4096 | 4096 |
| batch size | 64 | 128 | 128 |
| learning rate | 0.04 | 0.002 | 0.01 |
| optimizer | SGD | Adam | SGD |

Table 5: Hyperparameter settings for the Amazon Product Reviews dataset experiment for baselines

combinations according to various ranges for each hyper-parameter. For DIRT-T, the range of learning rate considered is 0.001-0.01, the range of dropout rate is 0.1-0.9. For MDD, the range of learning rate is [0.005,0.015], the range of dropout rate is 0.1-0.9 and the range of gamma is [1,2,3,4,5]. We try 5 different random seeds for each hyper-parameters combination and choose the best result.
We use the same dimensionality of the latent representation $Z$ for both baselines, and we used the same encoder architecture as used in the proposed method. The encoder architecture was selected for the proposed method and for these baselines according to the best performance obtained on the basic baseline DANN [3]. The dimensionality of the hidden representation $Z$ was selected among the values: 512, 1024, 2048, 4096 and the one with the best performance in DANN was retained.

## 4.4 Image Classification in the ImageCLEF Dataset

Here we present details regarding the experiment on the ImageCLEF dataset. For all baselines, we used reproduced results reported by previous studies.

- **The encoder** $\phi$ is a multi-layer perceptron (MLP) with a single hidden layer of 2048 dimensions.

- **The invariant representation** $Z$ is of 2048 dimensions.

- **The changing parameters** $\theta_X$ is implicitly calculated by concatenating the domain index $j$ to $X$ and $Z$ as input to the encoder and decoder respectively. For the decoder, $j$ is replicated the size of $Z$ times before being concatenated with $Z$.

- **The classifier** $h$ is an MLP of a single hidden layer with 2048 dimensions.

- **The decoder** $\tilde{\phi}$ is an MLP with a single of 2048 dimensions (which is the input dimensionality of the ImageCLEF data.).

- **The domain-adversarial discriminator** $h_a$ is an MLP of two hidden layer with 100 dimensions.

- **The regularization (auxiliary) classifier:** $h_c$ is an MLP with two hidden layers of 2048 dimensions each.

| Hyperparameter: | Value: |
|---|---|
| learning rate | 0.05 |
| $\lambda_\alpha$ | 0.01 |
| $\lambda_\gamma$ | 1 |
| $\lambda_\delta$ | 0.001 |
| batch size | 64 |
| optimizer | Adam |
| $L_2$ weight decay | 1.00E-04 |
| dropout | 0.70 |
| batch normalization | 0 |
| $\lambda_\tau$ | 1.0 |
| vada hyperparameter 1 | 1.0 |
| vada hyperparameter 2 | 0.5 |
| pseudo label hyperparameter 2 | 0.5 |

Table 6: Hyperparameter settings for the ImageCLEF dataset experiment.

## 5 Extended Discussion on Related Work

In this section, we briefly discuss the relationship between our work and other related methods for domain adaptation using neural architectures. In the main text, we mentioned recent advances in neural architectures such as spherical CNNs [5], source-target domain bridging via data augmentation [6] and pseudo-labels [10]. We note that these techniques do not mimic the data-generating process and ensure that all information is available to infer $Z$ and reconstruct the original data. However, the methods operate mainly on image datasets, and the techniques that they use are complementary to ours. We have attempted to incorporate some of them in our method, and we show in our ablation study that combining them with our approach is as a fruitful direction for improving performance in UDA. In this section, we focus on related studies that use domain-specific information when learning latent representations, in some shape or form.

### 5.1 Relationship to "Domain Separation Networks"[2]

"Domain Separation Networks" (DSN) is a study which is in principle most similar to ours. It relies on a autoencoder framework, in which there are two separate private encoders, in addition to a shared encoder, which together constitute in two separate encoding functions for the source and target domains, in order to infer $Z$. In addition to this, the authors explicitly model a private-part of $Z$

which can be thought of as analogous of our use of $\theta_X$ in our encoding and decoding functions, and the decoder is then a function of both the shared and private parts.

The main conceptual difference between this work and ours is that our work is motivated by the causal mechanism of generating $X$ from $\theta_X$ and $Z$. This inspired us to try to measure the influence of $\theta_X$ in this generating process (via the mutual information) and restrict it, thereby ensuring that $Z$ will not be a trivial and potentially useless representation for predicting $Y$. The method DSN achieves this as a by-product of using addition of the private and shared parts before providing them as input to the decoder which reconstructs the original features. This addition is a heuristic technique which is expected to specifically work with image datasets, whereas our mutual information minimization approach does not make such assumptions.

### 5.2 Relationship to ADDA [8]

The method ADDA is a well known domain adaptation technique which uses two separate encoders $\phi_S$ and $\phi_T$ to learn the latent representation $Z$. However, as mentioned in the main text, this is an inefficient way to parameterize the two separate encoding functions. Furthermore, this study lacks a decoder, and therefore does not attempt to regularize the latent representation $Z$ to have meaningful structure.

### 5.3 Relationship to "Continuously-Indexed Domain Adaptation" CIDA [9]

The method CIDA, like our method, takes domain index as input to the encoder $\phi$. However, in this method $\theta$ (the domain-specific information) is thought of a continuous random variable which is: **(1)**: of different value for each data-point, and **(2)**: has predictive information about the random variable $Y$. This reasoning assumes a different generating assumes a different generating process of the data than the one presented on Figure 1, in which $Y$ and $\theta$ are conditionally independent given $Z$. By assuming that $\theta$ has predictive information about $Y$, the authors implicitly assume that $\theta$ and $Y$ are dependent conditioned on $Y$ in the data-generating process, and thus their algorithm is tailored to benefit this information in $\theta$. On the other hand, our method only assumes that domain-specific information comes in the form of a discrete domain index unrelated to $Y$, and tries to make the best of that information to ensure the invariant representation $Z$ has sufficient quality.