# OpenReview forum: "Domain Adaptation with Invariant Representation Learning: What Transformations to Learn?"
_NeurIPS.cc/2021/Conference — NeurIPS 2021 Poster_

### Official Review · Reviewer_219a · 2021-07-11

**Rating:** 6
**Confidence:** 2

**Summary:**

The paper proposes a unsupervised domain adaptation method through the lens of learning invariant representations. The paper validates theoretically shows some limitations of current methods when learning from distinct source and target distributions. The paper validates experimentally on a large scale image dataset and 2 other datasets.

**Limitations And Societal Impact:**

The paper does not discuss any societal impacts.

**Main Review:**

Although the paper achieves promising results and the paper provides some theoretical justifications for their method, overall the paper is not motivated well and can be challenging to understand. For example, figure 2 is not explained very well in the text or caption, and figure 3 does not detail their proposed architecture. The architecture bears resemblance to Cycle-GAN and other related work. The paper cites CYCADA as related work but does not compare against it. The paper also does not cite very relevant work such as [1,2,3] and many others. The notation used can also be confusing in the text. The main novelty over CYCADA seems to be the L_{inv}^{trans}  which is used to minimized the KL divergence between marginals of the source and target distributions, however such ideas have existed in the community for some time.


[1] https://arxiv.org/abs/2004.07780
[2] https://openreview.net/forum?id=lQdXeXDoWtI
[3] https://arxiv.org/abs/2003.00688

**Time Spent Reviewing:**

2 hours

---

> ### Author Response · Authors · 2021-08-10
> **Response to the Reviewer's Questions and Concerns**
>
> We thank the reviewer for the comments and we address them below.
>
> Q1: “The architecture bears resemblance to Cycle-GAN and other related work”.
>
> A1: We respectfully disagree with this point. Our method does not have a cycle-consistency loss and it is not based on translation, but rather learning domain-invariant representations, to which we map both the source and the target domain observations. The translated data from source to target serves to \textit{regularize} the training of the invariant representation $Z$ and ensure that it preserves information for predicting $Y$ in the target domain. Learning domain-invariant representations has shown to be an effective framework for tackling the unsupervised domain adaptation problem, so it is not surprising that many studies rely on it. Our objective is to use this framework and improve it.
>
> Q2: “The paper cites CYCADA as related work but does not compare against it”
>
> A2: CYCADA is an image-translation based method which is typically run on digit classification tasks and on the original pixels of the image. Our method was not envisioned only for computer vision tasks, but rather diverse applications, as illustrated by the three diverse domain adaptation datasets that we used in our experiments. However, we are looking into your concern and attempting to run this baseline on ImageCLEF, and will update you as soon as possible.
>
> Q3: “The main novelty over CYCADA seems to be the L_{inv}^{trans} which is used to minimize the KL divergence between marginals of the source and target distributions, however such ideas have existed in the community for some time.”
>
> A3: Similarly to Cycle-GAN, CYCADA is also an image translation-based method which does not learn domain-invariant latent representations. Our proposed approach relies on an autoencoder framework and therefore it greatly differs from CYCADA in terms of the neural network architecture. Furthermore, $L_{inv}^{trans}$ is simple to implement but crucial for ensuring that the latent representation $Z$ is predictive of $Y$ in the target domain, as illustrated in our simulated experiments and supported by our real-world data experiments.
>
> Furthermore, we assume the reviewer meant that the  $L_{inv}^{trans}$ serves to minimize Jensen-Shannon divergence between the joint distributions, as presented in Theorem 1 and the explanation after it. We agree that the Jensen-Shannon Divergence is a classical way to measure a divergence between two probability distributions, and in itself is not novel, but to our knowledge it has never been employed to enforce minimal change of the joint distribution across domains in domain adaptation.
>
> We once again thank the reviewer for taking the time to review the manuscript, and we hope our response addressed the reviewer's concerns. We would be happy to answer any additional questions.

---

> > ### Comment · Reviewer_219a · 2021-08-23
> > **Thank you for your response**
> >
> > Thank you for your response.
> >
> > Q2: I agree with the authors.
> >
> > Q1 and Q3: thank you for clarifying the differences to CyCADA.
> >
> > I have adjusted my score.

---

### Official Review · Reviewer_8KhW · 2021-07-12

**Rating:** 8
**Confidence:** 3

**Summary:**

This paper analyzes the mechanism of domain adaptation and proposes a new strategy to reduce the gap between the two domains. The valuable information to achieve accurate classification can be well preserved. Extensive experiments are conducted on multiple dataset, such as Wi-Fi localization and Amazon Review.

**Limitations And Societal Impact:**

Yes

**Main Review:**

Advantages:

- It is interesting to understand the mechanism behind domain adaptation and this paper analyzes it from both empirical and theoretical perspectives. These discussions may bring some insights to the community.

- The experimental results show that the proposed method achieve higher performance compared with the existing methods, which verifies the effectiveness of the proposed method.

- This paper really make a contribution to the community.  Beyond the invariant features, the authors discuss that the domain-specifc information is also valuable. It has been empirically demonstrated by extensive experiments.

Concerns:
- The paper proposes a question about domain adaptation, i.e., 'What Transformations to Learn'. However, this question seems to not be answered explicitly. Could the authors conclude it with two or three sentences?
- The presentation of the paper can be refined.  For example, the fonts of captions in Figure 3 is very different from that in the main paper. The titles of Figure 4 can be improved.


**Time Spent Reviewing:**

7.5h

---

> ### Author Response · Authors · 2021-08-10
> **Response to the Reviewer's Questions and Concerns**
>
> We greatly appreciate the reviewer's feedback, and we address the reviewer's concerns below.
>
>
> Q1: The paper proposes a question about domain adaptation, i.e., 'What Transformations to Learn'. However, this question seems to not be answered explicitly. Could the authors conclude it with two or three sentences?
>
> A1: We appreciate the reviewer’s suggestion. In the update version of the manuscript, we shall reiterate that the transformations for feature learning in domain adaptation should take into account the data-generating process, and in our case, that means that one needs to ensure that the transformations take into account domain-specific information.
>
> Q2: “The presentation of the paper can be refined. For example, the fonts of captions in Figure 3 is very different from that in the main paper. The titles of Figure 4 can be improved.”
>
> A2: We agree with the reviewer and are grateful for the pointers. We will do our best to improve the presentation in the final version of the manuscript.
>
>
> We thank the reviewer once again for appreciating the merit of this work, and for providing suggestions and feedback.

---

> > ### Comment · Reviewer_8KhW · 2021-08-25
> > **Thank you for your response.**
> >
> > Thank you for your response. I will keep my score.

---

### Official Review · Reviewer_d96Z · 2021-07-13

**Rating:** 6
**Confidence:** 4

**Summary:**

**Summary and contributions:**
The paper considers representation learning for unsupervised domain adaptation (UDA). They show that, in some cases, it is necessary to learn encoders that vary across domains, something which is rarely done in practice. They thus propose a new method for incorporating domain-dependent features in the encoding process.

The end model/objective contains many different components: it is essentially a DANN with a decoder that aims to reconstruct the inputs by using domain-dependent and independent features. Furthermore, the decoder is forced to rely as little as possible on the domain-dependent features.


**Limitations And Societal Impact:**

Some limitations are briefly mentioned, e.g., that the method might not be as useful in high dimensions (line 177-181) "high dimension [...] no overlap [...] $\phi$only a function of X is sufficient". The discussion of limitations could be extended, e.g., the fact that the objective is very complicated and what are the downsides (e.g. hyperparameter tuning for balancing each component).

**Main Review:**

**Overall thoughts**: I think that the realization (and reasons) for using domain-dependent encoders are interesting and valuable. But I find the proposed objective too complicated (with 5 losses/components !)  to really be used by the community.

**Strengths**:
- **Valuable realization: domain-dependent encoders can be necessary**: The authors provide simple theoretical results that characterize some settings where domain-dependent encoders are necessary (e.g. conditional shift with shared support). Although very intuitive, I do not know of any previous work that pointed this out and I think that it is a valuable contribution to the UDA + representation learning subfield.
- **Useful/thorough appendices**: the appendices are useful and of good quality: extended related work, ablation, experimental settings… I especially appreciate the ablation study that investigates whether all the components of the final objective are necessary.
- **Decent performance**: On the presented datasets, the proposed method significantly outperforms the baselines. Similar results on standard UDA benchmarks would nevertheless be much stronger (see weaknesses below).

**Weaknesses**:
- **Very complicated objective**: I find the final objective to be a combination of too many terms (5 different components with their hyperparameters) to be practical. Ablation study shows that each component help, but there must be a simpler way of reaching the desired result.
- **Missing important benchmark**: the standard UDA benchmark for image classification is OfficeHome/Office31. These results are crucially missing. Even more so given that (1) the results on ImageCLEF seem strong; (2) you seem to provide code for OfficeHome (in `mdd.py`).


**Questions:**
- Important: how many seeds were used to generate the results from Fig. 4.b? Were the results averaged over all seeds or was the best selected? These are important experimental details that are missing. I found that information for other datasets but not ImageCLEF. Given that in the original table standard deviations were provided, please add standard deviations.
- Why do you bold the results from DSAN in table 4.b. when they are not the best (P->C or I->C)? I initially thought that the statistical difference might not have been significant but that wouldn’t explain why CDAN I->C is not bolded.
- I’m having difficulties understanding why the results are much lower for every model in the ablation study and somehow make a huge jump when everything is together. Did you use the same hyperparameter strategy for the ablation study? If so do you have an idea as to why that happens? Providing standard deviations would really help to make sense of those results.

**Suggestions**:
- explain somewhere that for ImageCLEF you are working directly from features from a Resnet50 pre-trained on ImageNet (if that is the case).
- use “section” instead of “chapter”
- define what you mean by “minimality”, currently, it’s simply thrown to you in line 277 without saying why this is what you need/meant.

**Minor suggestions**:
- line 21: $y^{\mathcal{S}}$ -> $y_k^{\mathcal{S}}$
- line 21: $x^{\mathcal{S}}$ -> $x_k^{\mathcal{S}}$
- line 49: domain -> domain
- line 50: .However -> . However
- line 83: $Z$ to contain -> $Z$ contains
- line 104: makes, -> makes <???>,
- line 108: of which -> of, which
- line 132: i.e. -> i.e.,
- line 132: low-dimensionality -> low information (? dimensionality has nothing fo with minimality)
- Proposition 1: I think you are pissing an assumption in the proposition that states that in the support the density is different (you say it in the text but it should appear in the proposition).
- Theorem 1: define $c_1,c_2$ (constants that upper bound [...])
- Figure 3: it’s currently hard to parse. Although it might have more to do with the complexity of the objective than the illustration. - I would also use a darker green to improve readability.
- Figure 4.b.: remove newline before “)”.
- Table 1: DANN t1-t2 ((1.3) -> (1.3)
- Table 1: what does parenthesis mean? if standard deviation then please use $\pm$
- Figure 5: withour -> without
- References: many duplicated entries e.g. 21-22 or 23-24 or 33-34. Please clean that up.
- Checklist: remove instructions (see line 486-487)



**Time Spent Reviewing:**

8 hours

---

> ### Author Response · Authors · 2021-08-10
> **Response to the Reviewer's Questions and Concerns**
>
> We greatly appreciate the reviewer’s detailed comments, suggestions, and concerns. We address the comments point-by-point below.
>
> Q1: “I find the final objective to be a combination of too many terms (5 different components with their hyperparameters) to be practical. Ablation study shows that each component help, but there must be a simpler way of reaching the desired result.”
>
> A1: We would like to note that unsupervised domain adaptation is a very challenging problem, and that our objective contains standard techniques (such as conditional entropy minimization and pseudolabels), in order to ensure that the invariant representation $Z$ has nontrivial structure, and they have been used in [1] (RSDA) and [2] (DIRT-T), among others. To our knowledge, these techniques have become standard in regularizing $Z$. Our addition to this set of regularizers are only the reconstruction loss of the decoder, and the classification loss of the auxiliary classifier which serves to minimize the Jensen-Shannon Divergence. We have observed that despite the five terms in the objective, our algorithm does not take significantly longer to train. We have been thinking about possible simpler ways to reach the desired result, but so far we haven’t got a clue.
>
> Q2: “Missing important benchmark: the standard UDA benchmark for image classification is OfficeHome/Office31. These results are crucially missing. “
>
> A2: We appreciate the reviewer’s concern about the absence of these results. We have run our method on pretrained ResNet features of this dataset, and provide it in the table below with several competitive and recent baselines for your reference. We label the baseline which uses pretrained ResNet features with “pretrained features” in parentheses, whereas we label the baselines which use ResNet finetuning as “ResNet finetune”. From the results, one can appreciate that even though our method does not statistically outperform the baselines which use ResNet finetuning, it is on par with them, despite the fact that it only relies on pretrained ResNet features. On the other hand, it does significantly outperform baselines that, like our method, make use of pretrained ResNet features.
>
> |                             | Art->Clipart | Art->Product | Art->RealWorld | Clipart->Art | Clipart->Product | Clipart->RealWorld | Product->Art | Product->Clipart | Product->RealWorld | RealWorld->Art | RealWorld->Clipart | RealWorld->Product | Average |
> |-----------------------------|--------------|--------------|----------------|--------------|------------------|--------------------|--------------|------------------|--------------------|----------------|--------------------|--------------------|---------|
> | MSTN [3] (pretrained features)  | 49.3         | 67.6         | 74.7           | 49.6         | 63.7             | 64.5               | 50.7         | 43.7             | 73                 | 62.7           | 52.2               | 77.9               | 60.9    |
> | DSR [4] (pretrained features)   | 53.4         | 71.6         | 77.4           | 57.1         | 66.8             | 69.3               | 56.7         | 49.2             | 75.7               | 68             | 54                 | 79.5               | 64.89   |
> | MDD [5] (ResNet finetune)       | 54.9         | 73.7         | 77.8           | 60           | 71.4             | 71.8               | 61.2         | 53.6             | 78.1               | 72.5           | 60.2               | 82.3               | 68.12   |
> | RSDA-DANN [1] (ResNet finetune) | 51.5+/-0.5   | 76.8+/-0.8   | 81.1+/-0.2     | 67.1+/-0.4   | 72.1+/-0.2       | 77.0+/-0.6         | 64.2+/-0.3   | 51.1+/-0.5       | 81.8+/-0.6         | 74.9+/-0.2     | 55.9+/-0.2         | 84.5+/-0.7         | 69.8    |
> | DSAN                        | 52.81+/-0.16 | 75.69+/-0.21 | 78.78+/-0.09   | 65.43+/-0.34 | 75.5+/-0.36      | 76.36+/-0.26       | 65.54+/-0.37 | 50.87+/-0.59     | 79.91+/-0.45       | 69.19+/-0.25   | 53.52+/-0.21       | 82.11+/-0.26       | 68.8    |
>
> Q3: “Important: how many seeds were used to generate the results from Fig. 4.b? Were the results averaged over all seeds or was the best selected? These are important experimental details that are missing. I found that information for other datasets but not ImageCLEF. Given that in the original table standard deviations were provided, please add standard deviations.”
>
> A3: For these results, we ran three random seeds and chose the best result among them. We provide mean and standard deviation from all random seed runs in the table below (please also see the response to the ablation experiment question).
>
> Q4: Why do you bold the results from DSAN in table 4.b. when they are not the best (P->C or I->C)?
>
> A4: We thank you for pointing it out and apologize for this mistake. We will correct it in the updated version of the manuscript.
>
> Q5: “I’m having difficulties understanding why the results are much lower for every model in the ablation study and somehow make a huge jump when everything is together. Did you use the same hyperparameter strategy for the ablation study? If so do you have an idea as to why that happens? Providing standard deviations would really help to make sense of those results.”
>
> A5: We used the same hyperparameter selection strategy for this study. Please find below the table of the ablation experiments with mean and standard deviation across three random seeds.
>
> | dec. | theta-enc. | theta-dec. | entropy | pseudo | reg. | i->p        | p->i        | i->c        | c->i        | c->p        | p->c        | average |
> |------|------------|------------|---------|--------|------|-------------|-------------|-------------|-------------|-------------|-------------|---------|
> | no   | yes        | N/A        | yes     | yes    | yes  | 78+/-0.9    | 92.4+/-2.8  | 95.7+/-0.6  | 92.77+/-0.9 | 75.03+/-1.4 | 92.77+/-0.4 | 87.77   |
> | yes  | yes        | no         | yes     | yes    | yes  | 79.15+/-0.2 | 92.96+/-0.6 | 96.44+/-0.1 | 93.61+/-0.5 | 78.22+/-1   | 91.44+/-2.9 | 88.63   |
> | yes  | no         | yes        | yes     | yes    | yes  | 79.01+/-0.5 | 93.22+/-0.6 | 96.33+/-0.2 | 93.28+/-0.4 | 77.33+/-2.7 | 94.49+/-0.7 | 88.94   |
> | yes  | yes        | yes        | no      | no     | yes  | 79.78+/-0.2 | 91.59+/-2.2 | 94.61+/-3   | 93.72+/-0.2 | 73.9+/-2.9  | 94.72+/-0.9 | 88.05   |
> | yes  | yes        | yes        | no      | yes    | yes  | 79.39+/-0.5 | 93.01+/-2   | 95.89+/-0.8 | 93.5+/-0.5  | 77.94+/-1.2 | 94.11+/-1.2 | 88.97   |
> | yes  | yes        | yes        | yes     | no     | yes  | 79.61+/-0.3 | 94.61+/-0.3 | 95.89+/-0.5 | 93.61+/-0.4 | 77.99+/-0.4 | 95.79+/-0.8 | 89.58   |
> | yes  | yes        | yes        | yes     | yes    | no   | 79.22+/-0.2 | 94.11+/-0.1 | 96.17+/-0.2 | 93.5+/-0.5  | 77.44+/-0.8 | 90+/-2.6    | 88.4    |
> | yes  | yes        | yes        | yes     | yes    | yes  | 80.33+/-0.5 | 95.23+/-0.3 | 97.54+/-0.2 | 94.66+/-0.5 | 79.61+/-0.6 | 96.67+/-0.2 | 90.07   |
>
> As observed from this table, we note that the increase in performance between the proposed method (including all its components, last row of the table) and the experiment without using pseudo-lables (third to last row of the table) is 1 %, whereas the experiment without using pseudo-labels is 2 % higher in accuracy than the experiment without using a decoder (first row), indicating that the jump in accuracy is not sudden. Overall, the goal of the ablation study is two-fold:
> 1) to demonstrate that all of the terms in our loss are important for the competitive performance  of the method, and
> 2)  to show that the performance is particularly poor when we do not take into account the likely data-generating process, which is the case in the first two rows of the table, in which no decoder is used, and/or no $\theta$ is provided to the decoder.
>
>
>
>
> We once again thank the reviewer for taking the time to provide the detailed feedback, and we hope our response addressed the reviewer's concerns. We would be happy to answer any further questions or concerns that the reviewer may have.
>
> References:
>
> [1] Gu, Xiang, Jian Sun, and Zongben Xu. "Spherical space domain adaptation with robust pseudo-label loss." Proceedings of the IEEE/CVF Conference on Computer Vision and Pattern Recognition. 2020.
>
> [2] Shu, Rui, et al. "A DIRT-T Approach to Unsupervised Domain Adaptation." International Conference on Learning Representations. 2018.
>
> [3] Xie, Shaoan, et al. "Learning semantic representations for unsupervised domain adaptation." International conference on machine learning. PMLR, 2018.
>
> [4] Cai, Ruichu, et al. "Learning disentangled semantic representation for domain adaptation." IJCAI: proceedings of the conference. Vol. 2019. NIH Public Access, 2019.
>
> [5] Zhang, Yuchen, et al. "Bridging theory and algorithm for domain adaptation." International Conference on Machine Learning. PMLR, 2019.

---

> > ### Comment · Reviewer_d96Z · 2021-08-18
> > **Comparison to RSDA-DANN**
> >
> > I have read all other reviews and your answers carefully. Thank you for your detailed answer. Especially for providing the OfficeHome comparison. I have a few questions concerning that:
> > 1. why are you comparing to RSDA-DANN instead of RSDA-MSTN from the same paper? The latter performs even better and significantly outperforms your method (average of 70.9). Similarly, why are you never comparing to RSDA-MSTN for other datasets in your paper (e.g. figure 4).
> > 2. Concerning this distinction about fine-tuned/pretrained features, why is that not mentioned in the paper ? Specifically, in figure 4 are you using finetuned or pretrained ? what about RSDA-DANN in figure 4 ?

---

> > > ### Author Response · Authors · 2021-08-20
> > > **Respons to the Follow-Up Questions by the Reviewer**
> > >
> > > Dear Reviewer,
> > >
> > > Thank you very much for carefully reading our response and the reviews. We greatly appreciate your informative comments and insights. Below please find our detailed response, along with more experimental results:
> > >
> > > Regarding your first question: "why are you comparing to RSDA-DANN instead of RSDA-MSTN from the same paper? The latter performs even better and significantly outperforms your method (average of 70.9). Similarly, why are you never comparing to RSDA-MSTN for other datasets in your paper (e.g. figure 4).": thank you very much for noticing that result was missing in Figure 4b). In light of your comment, we have included RSDA-DANN and RSDA-MSTN with and without finetuning in a revised table of the ablation experiment from our previous response, which is also provided below. We will provide more discussion about these results, along with your concern regarding the Office-Home results, in the response to your second question.
> > >
> > > Furthermore, one may notice that the RSDA approach is orthogonal to ours, and that its main contribution is a different type of neural network architecture to use in the invariant feature learning framework such as DANN. In the specific case of RSDA, the feature map from X to Z (the invariant representation) is a spherical neural network. We conjecture that this framework can be combined with ours to further improve the performance. We are currently investigating this direction and will update you on the experimental results as soon as possible.
> > >
> > > Regarding your second question: "Concerning this distinction about fine-tuned/pretrained features, why is that not mentioned in the paper ? Specifically, in figure 4 are you using finetuned or pretrained ? what about RSDA-DANN in figure 4 ?": thanks a lot for being thoughtful and appreciating this core difference in the training settings. The experiments for the proposed method on ImageCLEF were run on pretrained ResNet features, and in Fig. 4b the reported results for baselines are the same results that were reported in the original work, so the use of fine-tuning depends on how each respective study was originally conducted. This means that for RSDA-DANN and RSDA-MSTN, the reported result was obtained with finetuning. As mentioned in the answer to your previous question, to further elucidate the effect of finetuning on the performance, we conducted experiments on RSDA-DANN and RSDA-MSTN on the same pre-trained ResNet features that we used for our method. The table is presented below (the last row corresponds to our proposed method with all of the loss terms included):
> > >
> > > | dec. | theta-enc. | theta-dec. | entropy | pseudo | reg. | i->p        | p->i        | i->c        | c->i        | c->p        | p->c        | average |
> > > |------|------------|------------|---------|--------|------|-------------|-------------|-------------|-------------|-------------|-------------|---------|
> > > | no    | yes           | N/A           | yes       | yes      | yes   | 78+/-0.9    | 92.4+/-2.8  | 95.7+/-0.6  | 92.77+/-0.9 | 75.03+/-1.4 | 92.77+/-0.4 | 87.77   |
> > > | yes   | yes           | no             | yes        | yes     | yes   | 79.15+/-0.2 | 92.96+/-0.6 | 96.44+/-0.1 | 93.61+/-0.5 | 78.22+/-1   | 91.44+/-2.9 | 88.64   |
> > > | yes   | no            | yes            | yes        | yes     | yes   | 79.01+/-0.5 | 93.22+/-0.6 | 96.33+/-0.2 | 93.28+/-0.4 | 77.33+/-2.7 | 94.49+/-0.7 | 88.94   |
> > > | yes   | yes           | yes            | no         | no      | yes   | 79.78+/-0.2 | 91.59+/-2.2 | 94.61+/-3   | 93.72+/-0.2 | 73.9+/-2.9  | 94.72+/-0.9 | 88.05   |
> > > | yes   | yes           | yes            | no         | yes     | yes   | 79.39+/-0.5 | 93.01+/-2   | 95.89+/-0.8 | 93.5+/-0.5  | 77.94+/-1.2 | 94.11+/-1.2 | 88.97   |
> > > | yes   | yes           | yes            | yes        | no      | yes   | 79.61+/-0.3 | 94.61+/-0.3 | 95.89+/-0.5 | 93.61+/-0.4 | 77.99+/-0.4 | 95.79+/-0.8 | 89.58   |
> > > | yes   | yes           | yes            | yes        | yes     | no    | 79.22+/-0.2 | 94.11+/-0.1 | 96.17+/-0.2 | 93.5+/-0.5  | 77.44+/-0.8 | 90+/-2.6    | 88.4    |
> > > RSDA-DANN (with ResNet finetuning)  |N/A|N/A|N/A|N/A|N/A| 79.20+/-0.4 | 93.0 +/-0.2  | 98.3+/-0.4   | 93.6+/-0.4  |78.5+/-0.3    | 98.2+/-0.2 | 90.1   |
> > > RSDA-DANN (with pretrained features) |N/A|N/A|N/A|N/A|N/A| 76.5 +/-0.2 | 86.06+/-0.4 | 94.44+/-0.8  | 92.55+/-0.5|78.0+/-0.2    |89.0+/-0.1  |   86.1 |
> > > RSDA-MSTN (with ResNet finetuning)    |N/A|N/A|N/A|N/A|N/A| 79.8+/-0.4   | 94.5+/-0.5   | 98.0+/-0.4   | 94.2 +/-0.4 | 79.2+/-0.3   | 97.3+/-0.3 | 90.5   |
> > > RSDA-MSTN (with pretrained features)    |N/A|N/A|N/A|N/A|N/A| 76.39+/-0.3 | 89.39+/-0.2 | 95.89+/-0.7|89.28+/-0.3  | 76.5+/-0.1   | 92.94+/-0.2| 86.73|
> > > | yes   | yes           | yes            | yes        | yes     | yes   | 80.33+/-0.5 | 95.23+/-0.3 | 97.54+/-0.2 | 94.66+/-0.5 | 79.61+/-0.6 | 96.67+/-0.2 | 90.7   |
> > >
> > > From the table, one can appreciate that when comparing our approach to RSDA-MSTN and RSDA-DANN trained on the same pretrained features as our method, our method outperforms these baselines even more convincingly. Therefore, a likely reason for the edge that the published RSDA-MSTN approach has over our method in Office-Home is the difference in training settings (the fact that this baseline used fine-tuning for the published result, while our method used pre-trained features). We are currently running RSDA-MSTN and RSDA-DANN experiments on pre-trained features on the Office-Home dataset, and we will update you within the next 48 hours.
> > >
> > > Finally, we would like to add that the reason why we used pretrained features in the experiments of our method is because it is more difficult to learn to reconstruct the original high-dimensional image in these particular benchmark datasets than the pretrained features, given the limited amount of training data.
> > >
> > > Please let us know if you have any further questions or concerns.

---

> > > > ### Author Response · Authors · 2021-08-24
> > > > **Thanks for your patience; we are still running experiments and will update you shortly**
> > > >
> > > > Dear Reviewer d96Z,
> > > >
> > > > We apologize that we have not provided the results as promised before. Thank you very much for your patience. We are writing to let you know that we are still running the experiments, and will update you shortly.
> > > >
> > > > Thanks for your understanding.

---

> > > > > ### Author Response · Authors · 2021-08-24
> > > > > **Follow-Up Experiment Results**
> > > > >
> > > > > Dear Reviewer d96Z,
> > > > >
> > > > > We are writing to provide results for the RSDA-MSTN and RSDA-DANN baselines run on the Office-Home dataset, when using pretrained features instead of finetuning. The results are shown in the table below:
> > > > >
> > > > > | Art->Clipart | Art->Product | Art->RealWorld | Clipart->Art | Clipart->Product | Clipart->RealWorld | Product->Art | Product->Clipart | Product->RealWorld | RealWorld->Art | RealWorld->Clipart | RealWorld->Product | Average      |       |
> > > > > |--------------|--------------|----------------|--------------|------------------|--------------------|--------------|------------------|--------------------|----------------|--------------------|--------------------|--------------|-------|
> > > > > | RSDA-DANN    | 49.48+/-1.45 | 67.55+/-1.68   | 74.99+/-0.2  | 59.03+/-2.01     | 72.7+/-0.76        | 72.95+/-1.36 | 63.36+/-1.37     | 47.77+/-1.3        | 82.05+/-0.94   | 67.24+/-1.19       | 53.49+/-1.48       | 84.35+/-0.2  | 66.24 |
> > > > > | RSDA-MSTN    | 51+/-1.08    | 72.37+/-1.05   | 75.48+/-0.68 | 63.22+/-1.27     | 75+/-0.67          | 73.54+/-1.82 | 66.1+/-1.5       | 49.81+/-0.3        | 80.01+/-0.49   | 70.22+/-0.58       | 53.93+/-0.24       | 82.65+/-1.3  | 67.77 |
> > > > > | DSAN         | 52.80+/-0.16 | 75.69+/-0.21   | 78.78+/-0.09 | 65.43+/-0.34     | 75.5+/-0.36        | 76.36+/-0.26 | 65.54+/-0.37     | 50.87+/-0.59       | 79.91+/-0.45   | 69.19+/-0.25       | 53.52+/-0.21       | 82.11+/-0.26 | 68.8  |
> > > > >
> > > > > From this table, one can appreciate that the proposed method outperforms the baselines when all methods are run on pretrained features.
> > > > >
> > > > > We apologize again for the delays in providing this result, and we thank you for your patience and understanding. Please let us know if you have any additional questions or concerns.

---

### Official Review · Reviewer_DkiW · 2021-07-14

**Rating:** 6
**Confidence:** 4

**Summary:**

This paper focuses on making use of latent invariant representations for unsupervised domain adaptation. The goal is to address two State-Of-The-Art existing methods issues. The first is there is no principled way to ensure that marginal invariance (P^S(Z) = P^T (Z)) preserves conditional invariance (P^S (Z|Y) = P^T (Z|Y)). The second is there are cases (especially in lower-dimensional datasets) where the encoding function Phi needs domain-specific information, in addition to the features X.
This paper introduces an invariant representation learning method that uses the data-generating process to justify and efficiently use domain-specific information in the encoder Phi and constraint it to ensure that Z contains relevant label-specific information.
Results on three real datasets and one synthetic dataset show the proposed Domain-Specific Adversarial Network (DSAN) model’s performances (accuracy and Wilcoxon p-value) compared with existing methods and an unregularized DSAN (DSAN-U) version.


**Ethical Concerns:**

There is no ethical issue in this paper.

**Limitations And Societal Impact:**

The authors write no elements about limitations and societal impact.

**Main Review:**

#Originality:

Using the data-generating process under conditional shift based on the causal mechanism of generating X from Z with Theta_X represents the changing parameters of P(X|Y) across domains is a brilliant idea. I appreciated how the authors motivated the use of two separate encoders to infer Z in chapter 2. Indeed, model Theta_X as a latent variable and provide it as an input to Phi in addition to X and, then, mimic the causal mechanism generating X via a decoder constraining the influence of Theta_X to be minimal (acting as a regularizer on the first encoder and forcing it to preserve semantic information when inferring Z) is very interesting and well intuitively explained.

#Quality:

The submission technically sounds well, but there are mathematical notation problems: in line 21, k and m_S, m_T are not defined. Line 152, in proposition 1, the output space of Phi should be corrected. The notation of A elements must be homogenized (we meet a x in proposition 1 and an a in proposition 2). In proposition 2 it could be more appropriated to put the indices risks such as \epsilon_S^A(g) instead of \epsilon ^A(g) _S. Line 217 the second “reconstruct” loss has to be modified by “classif” loss. In Theorem 1 some comas are missed in distribution formulations. After line 256, there are some mistakes in the equations: the index j seems to be corrected by c. Line 259 denoted \lambda_T could be confusing notation because T is yet used for the target domain. After line 260, in the equations, \rho and h_a are not defined.
It could be appreciated that authors give more intuitions of some concepts such as “Markov Blanket” (line 141). Also, the authors have to improve the end of chapter 3 because it is not intuitively and difficult to follow.
Then, in line 210, authors must cancel one “label” word. Line 253, “RHS” is not defined.
To conclude, this work seems to be in progress and should be improved because of all these points.

#Clarity:

Please, read the previous remarks about “Quality” to understand why the submission is not clearly written.

#Significance:

The experimental results sound well bust MSTN approach is announced line 312 but missing in tables. Also, in line 326, the authors have to rectify the Figure 4(b) reference instead of “Table 4(b) because there is no table 4 in the paper.



**Time Spent Reviewing:**

Around 5 hours.

---

> ### Author Response · Authors · 2021-08-10
> **Response to the Reviewer's Questions and Concerns**
>
> We are very grateful to the reviewer for the detailed comments and suggestions. We address the comments point-by-point below.
>
> Q1: “The submission technically sounds well, but there are mathematical notation problems: in line 21, k and m_S, m_T are not defined.”
>
> A1: Thanks for spotting it. They represent the number of observations in the source and target domains respectively. We will add this definition to the updated version of the manuscript.
>
> Q2: “Line 152, in proposition 1, the output space of Phi should be corrected. The notation of A elements must be homogenized (we meet a x in proposition 1 and an a in proposition 2). In proposition 2 it could be more appropriated to put the indices risks such as \epsilon_S^A(g) instead of \epsilon ^A(g) _S.”
>
> A2: We greatly appreciate these suggestions in fixing notation and making the theoretical statements clearer. We will incorporate them in the updated version of the manuscript.
>
> Q3: “Line 217 the second “reconstruct” loss has to be modified by “classif” loss. In Theorem 1 some comas are missed in distribution formulations. After line 256, there are some mistakes in the equations: the index j seems to be corrected by c. Line 259 denoted \lambda_T could be confusing notation because T is yet used for the target domain.”
>
> A3: We again thank the reviewer for taking note of these issues in notation, and will fix them and incorporate the changes in the updated version of the manuscript.
>
> Q4: “What is the meaning of $\rho$ and $h_a$?”
>
> A4: $\rho$ is a function that maps the domain index into a hidden vector $\theta_X$ which is then concatenated with $X$ and provided to the encoder $\phi$. Similarly, it is concatenated with $Z$ and provided to the decoder $\tilde{\phi}$. Furthermore, $h_a$ corresponds to the domain-adversarial predictor that predicts the domain index from $Z$ (as implicitly introduced in line 212), in order to ensure that $Z$ is domain-invariant.  We will explicitly define these terms in the second paragraph of section 3 in the updated version of the manuscript.
>
> Q5: “The users should define the meaning of Markov Blanket”
>
> A5: For a particular feature $Y$, and given a directed acyclic probabilistic graphical model, the Markov Blanket represents the feature's parents, the feature’s children, and the parents of the feature's children in the graph. $Y$ is conditionally independent of all of the other features, given the features in its Markov Blanket. Therefore, the only features that are relevant for predicting $Y$ are the ones in its Markov Blanket. We will incorporate the definition (as found in standard textbooks in graphical models) in the updated version of the manuscript.
>
> Q6: “The experimental results sound well but MSTN approach is announced line 312 but missing in tables.”
>
> A6: We appreciate the reviewer's concern, but we would like to note that the MSTN approach is in fact present in the ImageCLEF results on Figure 4(b).
>
> We thank the reviewer once again for reviewing our manuscript, and we hope our response provides the needed clarifications. We would be happy to answer any further questions the reviewer may have.

---

> > ### Comment · Reviewer_DkiW · 2021-08-23
> > **Thank you for your response and clarifications**
> >
> > Thank you for your response and clarifications. Following the authors' engagements to update the manuscript, I have adjusted my score.

---

> > > ### Author Response · Authors · 2021-08-24
> > > **Thank you for the update**
> > >
> > > We are grateful for your consideration of our response and the update of your recommendation!

---

### Decision · Program_Chairs · 2021-09-27

**Decision:**

Accept (Poster)

**Comment:**

The authors study unsupervised domain adaptation. They point out that if the data supports overlap, the encoder might need domain-specific information to recover an invariant representation which elicits a high-accuracy classifier. They propose a method which models this domain-specific information as a latent variable while encouraging it to have minimal influence on a decoder.

Reviewers agreed that the motivating analysis (necessity of domain-dependent encoders) was informative. Most reviewers (DkiW, 8KhW, 219a) thought the experiments were promising (reviewer d96Z remarked that OfficeHome results were missing, but the authors address this in the rebuttal). Reviewers also tended to agree that the method itself was overcomplicated and this might hurt its usefulness in practice (319a, d96Z), but I feel this shouldn't by itself prevent acceptance.